# Piwi–piRNA complexes induce stepwise changes in nuclear architecture at target loci

Yuka W Iwasaki[1,2,*] iD, Sira Sriswasdi[3,4,5,†] iD, Yasuha Kinugasa[6,†] iD, Jun Adachi[7] iD, Yasunori Horikoshi[6], Aoi Shibuya[1], Wataru Iwasaki[3] iD, Satoshi Tashiro[6], Takeshi Tomonaga[7] & Haruhiko Siomi[1,**] iD

## Abstract

PIWI-interacting RNAs (piRNAs) are germline-specific small RNAs that form effector complexes with PIWI proteins (Piwi–piRNA complexes) and play critical roles for preserving genomic integrity by repressing transposable elements (TEs). *Drosophila* Piwi transcriptionally silences specific targets through heterochromatin formation and increases histone H3K9 methylation (H3K9me3) and histone H1 deposition at these loci, with nuclear RNA export factor variant Nxf2 serving as a co-factor. Using ChEP and DamID-seq, we now uncover a Piwi/Nxf2-dependent target association with nuclear lamins. Hi-C analysis of Piwi or Nxf2-depleted cells reveals decreased intra-TAD and increased inter-TAD interactions in regions harboring Piwi–piRNA target TEs. Using a forced tethering system, we analyze the functional effects of Piwi–piRNA/Nxf2-mediated recruitment of piRNA target regions to the nuclear periphery. Removal of active histone marks is followed by transcriptional silencing, chromatin conformational changes, and H3K9me3 and H1 association. Our data show that the Piwi–piRNA pathway can induce stepwise changes in nuclear architecture and chromatin state at target loci for transcriptional silencing.

**Keywords** chromatin conformation; heterochromatin formation; nuclear localization; RNA silencing; transcriptional regulation
**Subject Categories** Chromatin, Transcription & Genomics; RNA Biology
**The EMBO Journal (2021) 40: e108345**

## Introduction

Gene expression can be modulated by subnuclear localization (van Steensel & Furlong, 2019). Representatively, heterochromatin segregates spatially from euchromatin and is localized preferentially toward the nuclear periphery and surrounding the nucleolus (Pombo & Dillon, 2015). Indeed, gene distribution relative to the nuclear lamina influences both chromatin organization and gene regulation (Gonzalez-Sandoval & Gasser, 2016; van Steensel & Belmont, 2017). Genome-wide detection of regions interacting with the nuclear lamina is possible using DamID (Pickersgill *et al*, 2006). Such analyses have revealed lamina-associated domains (LADs) with specific chromosomal characteristics and gene regulatory functions (Harr *et al*, 2016). Recent studies in mouse ES cells and *Drosophila* have shown that nuclear lamin depletion alters chromatin conformation and gene expression within specific chromatin regions (Zheng *et al*, 2018; Ulianov *et al*, 2019). Artificial tethering of chromatin regions to the nuclear periphery has revealed the impact of lamins on transcription, but regions tethered to lamins do not uniformly show transcriptional silencing (Finlan *et al*, 2008; Kumaran & Spector, 2008; Reddy *et al*, 2008). Whether a chromatin region is affected or not depends on attributes including histone modifications and is highly context-dependent. Little is known about the mechanisms underlying this context dependence.

At a finer level, the dynamic three-dimensional (3D) structure of the genome is essential for a range of biological functions (Bonev & Cavalli, 2016). Genome organization is dynamic, but non-random, comprising transcriptionally active and inactive compartments called A/B compartments (Lieberman-Aiden *et al*, 2009) that further sub-divide into topologically associating domains (TADs) with high intra-domain interactions and low inter-domain interactions (Dixon *et al*, 2012; Nora *et al*, 2012). High-throughput chromosome conformation capture, Hi-C, has revealed that the *Drosophila* genome is organized into TADs bounded by insulator complex proteins, and chromatin state and genome structure are linked at both TAD and compartment levels (Hou *et al*, 2012; Sexton *et al*, 2012). There is mounting evidence that LADs and TADs can function together as

1  Department of Molecular Biology, Keio University School of Medicine, Tokyo, Japan
2  Japan Science and Technology Agency (JST), Precursory Research for Embryonic Science and Technology (PRESTO), Saitama, Japan
3  Department of Biological Sciences, Graduate School of Science, The University of Tokyo, Tokyo, Japan
4  Computational Molecular Biology Group, Faculty of Medicine, Chulalongkorn University, Bangkok, Thailand
5  Research Affairs, Faculty of Medicine, Chulalongkorn University, Bangkok, Thailand
6  Department of Cellular Biology, Research Institute for Radiation Biology Medicine, Hiroshima University, Hiroshima, Japan
7  Laboratory of Proteome Research, National Institutes of Biomedical Innovation, Health and Nutrition, Osaka, Japan
  *Corresponding author. Tel: +81 3 5363 3529; E-mail: iwasaki@keio.jp
  **Corresponding author. Tel: +81 3 5363 3754; E-mail: awa403@keio.jp
  †These authors contributed equally to this work

genome organizers (Paulsen *et al*, 2017; Borsos *et al*, 2019). However, the molecular mechanisms that act to regulate specific loci in the context of LADs and TADs remain unknown.

The PIWI-interacting RNA (piRNA) pathway preserves genome integrity by repressing transposable elements (TEs) in animal germ cells (Iwasaki *et al*, 2017; Ozata *et al*, 2019). *Drosophila* Piwi forms effector complexes with piRNAs to transcriptionally silence its targets through heterochromatin formation, which in turn regulates cofactors including H3K9me3 methyltransferase Eggless/SETDB1 (Egg), HP1a, and linker histone H1 (Sienski *et al*, 2012; Le Thomas *et al*, 2013; Rozhkov *et al*, 2013; Sienski *et al*, 2015; Yu *et al*, 2015; Iwasaki *et al*, 2016). Importantly, it has been shown that Piwi–piRNA-mediated silencing decreases chromatin accessibility, suggesting dynamic regulation of chromatin upon target TE regulation (Iwasaki *et al*, 2016). Recently, it has been identified that the Piwi–Panx–Nxf2–p15 (PPNP a.k.a. Pandas, PICTS, SFiNX) complex plays a central role in bridging target-associated Piwi–piRNA to chromatin regulation and transcriptional silencing (Batki *et al*, 2019; Fabry *et al*, 2019; Murano *et al*, 2019; Zhao *et al*, 2019). Nxf2, a nuclear RNA export factor (NXF) variant, interacts with Piwi as a member of the PPNP complex and plays an essential role in Piwi–piRISC-mediated silencing. Nxf2 associates with p15/Nxt1, a co-adaptor for nuclear RNA export which interacts with the nuclear pore complex (Fribourg *et al*, 2001; Herold *et al*, 2001; Kerkow *et al*, 2012). Nxf2–p15 functions in transposon silencing, unlike Nxf1–p15 which plays a major role in mRNA export. How Nxf2–p15 mediates silencing in the Piwi–piRNA pathway, rather than RNA export, and whether this complex impacts nuclear localization of target RNAs remain unclear. Moreover, whether Piwi–piRNA and its silencing complex, which associate with nascent RNA rather than chromatin, regulate chromatin conformation to trigger heterochromatin formation is unknown. Recent studies indicate that silencing complexes formed by Panx–Nxf2–p15 further associate with the dynein light chain protein, LC8, and form condensates in a dimer-dependent manner to induce co-transcriptional silencing (Eastwood *et al*, 2021; Schnabl *et al*, 2021). However, whether Piwi–piRNA regulation impacts nuclear spatial organization *in vivo* is still unrevealed.

Here, using ovarian somatic cells (OSCs), we performed chromatin enrichment for proteomics (ChEP) analysis, a technique capable of detecting global chromatin-associated protein composition (Kustatscher *et al*, 2014). ChEP analysis showed a decrease in chromatin-associated lamins (Lamin and Lamin C) upon Piwi depletion. Lamin DamID-seq analysis revealed that TEs regulated by Piwi–piRISCs interact with the nuclear lamina and become detached upon Piwi or Nxf2 knockdown. This suggests that Nxf2–p15 may recruit Piwi–piRNA targets to the nuclear periphery upon transcriptional silencing. Hi-C analysis demonstrated that Piwi depletion does not affect overall TAD structure, but results in decreased intra-TAD and increased inter-TAD interactions. Interestingly, changes in active histone marks are altered with changes in nuclear localization, while repressive histone marks associated with chromatin conformational change. Forced tethering of Nxf2 to the nascent reporter transcript caused localization of the reporter coding locus to the nuclear periphery along with initiation of transcriptional regulation. These results suggest that regulation of TEs by Piwi–piRNAs is initiated by changes in nuclear localization and removal of active histone marks and then followed by impacts on chromatin

conformation and repressive histone marks. These stepwise changes further lead to the formation of stable heterochromatin structure.

# Results

## Piwi–depletion results in a decrease in chromatin-associated lamins

To identify chromatin factors impacted by Piwi–piRNA regulation, we performed chromatin enrichment for proteome (ChEP; Kustatscher *et al*, 2014) analysis using ovarian somatic cells (OSCs), a cultured ovarian somatic cell line in which piRNA-guided silencing operates (Niki *et al*, 2006; Saito *et al*, 2009). OSCs were treated with either EGFP-targeting (control) or Piwi-targeting siRNAs for RNAi knockdown (KD; Fig EV1A). ChEP analysis can detect global chromatin-associated protein composition (Kustatscher *et al*, 2014). Therefore, by comparing ChEP data obtained from Control- or Piwi-KD OSCs, we can detect proteins with altered chromatin association levels. This approach identified 25 candidate proteins whose chromatin association was affected upon Piwi-KD (either increased or decreased, adjusted *P*-value < 0.05) (Fig 1A; Table EV1). Linker histone H1, which is known to dissociate from chromatin upon Piwi-KD (Iwasaki *et al*, 2016), was also included in the list of extracted proteins. Gene ontology analysis of 25 candidate proteins revealed that these proteins were linked to chromatin silencing and DNA binding, as expected (Fig 1B). Additionally, terms such as nuclear pore distribution, intermediate filament, and lamin filament were extracted. Closer inspection of the genes listed under these terms revealed that both *Drosophila* Lamin and LaminC are included (Figs 1A and EV1B).

Knockdown of Piwi did not affect Lam or LamC expression levels (Fig 1C) or localization of lamin proteins to the nuclear periphery (Fig 1D). Additionally, interaction between Piwi and lamin proteins was not detectable (Fig EV1C). However, Lam- or LamC-KD resulted in a slight de-repression of piRNA target TEs (Fig 1E), suggesting that accession to the nuclear periphery may impact TE repression to some extent. Due to severe damage to Lam- or LamC-KD cells, we could only perform knockdown for 48 h, which cannot completely deplete their expression (Fig 1C). Similarly, double KD of Lam and LamC proteins severely affected cell viability. Incomplete KD may result in the underestimation of its effect on TE de-repression upon lamin loss. These results together suggest that Piwi does not affect Lamin or LaminC expression or localization. Instead, Piwi-KD results in decreased association of chromatin regions with the nuclear periphery, and this may have some impact on the regulation of piRNA target TEs.

## Piwi–piRNA target TE regions are tethered to the nuclear periphery in a Piwi-dependent manner

It is unclear whether the decrease in the association of chromatin with lamins, as indicated by ChEP analysis, is linked to Piwi–piRNA regulation or global changes in cellular conditions upon Piwi-KD. To investigate this, we next analyzed which chromatin regions lose association with the nuclear periphery or lamins upon Piwi-KD, using DamID-seq analysis of Lamin in two replicates (Fig EV1D). Dam-fused Lamin protein methylates proximal

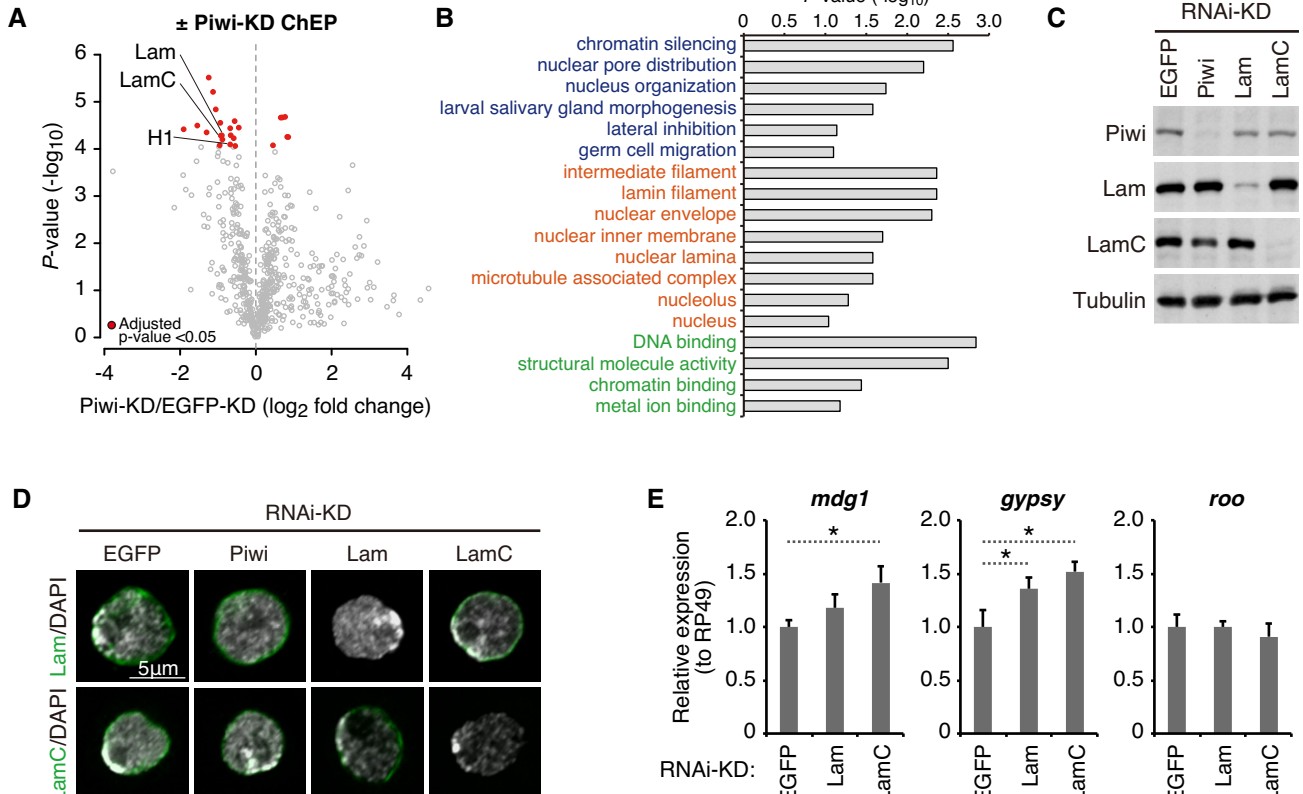

**Figure 1. Piwi–depletion results in decreased chromatin-associated Lamins.**

A   ChEP analysis using EGFP (control)- or Piwi-KD OSCs. Signals with an adjusted *P*-value (Bonferroni test) of < 0.05 are highlighted in red. The signals for Lam, LamC, and H1 are labeled within the plots.

B   Gene ontology terms related to proteins with a significant change upon Piwi-KD using ChEP analysis (adjusted *P*-value < 0.05). Gene ontology terms from the biological process (blue), cellular component (orange), and molecular function (green) categories are listed along with their *P*-values (Fisher's exact test, −log$_{10}$).

C   Western blot showing Piwi, Lam, LamC, and Tubulin (loading control) protein levels upon EGFP (control)-, Piwi-, Lam-, or LamC-KD. Note that KD of Lam or LamC is not complete since their depletion causes serious cell damage.

D   Immunofluorescence of OSCs transfected with indicated siRNAs, using Lam and LamC antibodies (green). DAPI staining (gray) shows the location of nuclei. Scale bar: 5 μm.

E   Lam- or LamC-KD followed by qRT–PCR of RNA levels for piRNA target (*mdg1* and *gypsy*) or non-target (*roo*) TEs. Error bars indicate SD (*n* = 3). *P*-value < 0.05 (*t*-test). Note that the complete depletion of Lam and LamC is impossible since their KD causes severe damage to cells.

Source data are available online for this figure.

chromatin regions, and the resulting methylated regions can be digested by restriction enzyme (Pickersgill *et al*, 2006). Deep sequencing can then identify regions that were associated with Lamins. The ratio of the resulting reads over background (reads obtained under Dam expression alone) indicates the proximity of sequences to the nuclear lamin. We first compared Lamin DamID-seq signals from Control- and Piwi-KD samples, for FlyBase-annotated genes and transposons (Fig 2A). This analysis revealed that though the DamID signal for most FlyBase genes remained consistent upon Piwi-KD, the signal decreased in certain populations of transposons. For example, *gypsy* insertion at chromosome 2R showed the biggest decrease in DamID signal upon Piwi-KD. A genome browser view of the genomic region with this *gypsy* insertion indicates a clear decrease of Lamin DamID signal, whereas the region with *gypsy10* and *accord* insertion, which are not targeted by piRNAs, was unaffected (Fig 2B).

To analyze whether the difference in DamID signal observed among TE types is linked to Piwi–piRNA regulation, we analyzed the expression level and piRNA targeting of TEs by mapping RNA-seq and small RNA-seq reads to TE consensus sequences (Iwasaki *et al*, 2016; Data ref: Iwasaki *et al*, 2016). piRNA target TEs which had over 10-fold de-repression upon Piwi-KD were labeled as "piRNA target TEs" (Figs 2C and EV1E). We could confirm that Piwi-associated piRNAs were mapped specifically to theses TEs (Figs 2C and EV1F). Lamin DamID signals for "piRNA target TEs" decreased upon Piwi-KD, whereas signals for "non-piRNA target TEs" remained unchanged (Fig 2D and E). As shown for the *gypsy* and *mdg1* piRNA target TEs, loss of Piwi resulted in a decrease of Dam-associated signal from a positive to negative value. This was not observed for non-piRNA target TEs such as *roo* (Fig 2F).

Chromatin regions with positive Lamin DamID-seq signals are defined as lamina-associated domains (LADs; Pickersgill *et al*, 2006;

van Bemmel *et al*, 2010). LADs are associated with components of the nuclear envelope that tether them to the nuclear periphery. We further defined LADs using Lamin DamID-seq data from the Control-KD sample and identified 483 LADs. Analysis of changes in DamID signal per LAD upon Piwi-KD identified decreased DamID signals specifically for LADs harboring piRNA target TEs (Fig 2G). This indicates that LADs harboring piRNA target TEs have decreased contact with Lamins upon Piwi-KD. In order to confirm this, we designed a probe targeting the genomic regions around *gypsy* insertion at chromosome 2L and performed oligonucleotide fluorescence *in situ* hybridization (oligo-FISH) analysis (Fig 2H). We measured the distance between the DAPI surface and the FISH

signal (Fig EV1G) and found a significant increase in distance from the nuclear periphery to FISH signal upon Piwi-KD. Together, these results suggest that Piwi-KD specifically causes detachment of piRNA target TEs from the nuclear periphery (Fig 2I). This suggests that Piwi can recruit piRNA target genes to the nuclear periphery upon transcriptional silencing.

## Decreased intra-TAD interactions and increased inter-TAD interactions upon Piwi–depletion

It has been reported in several species, including *Drosophila*, that localization of chromatin regions to the nuclear periphery or LADs

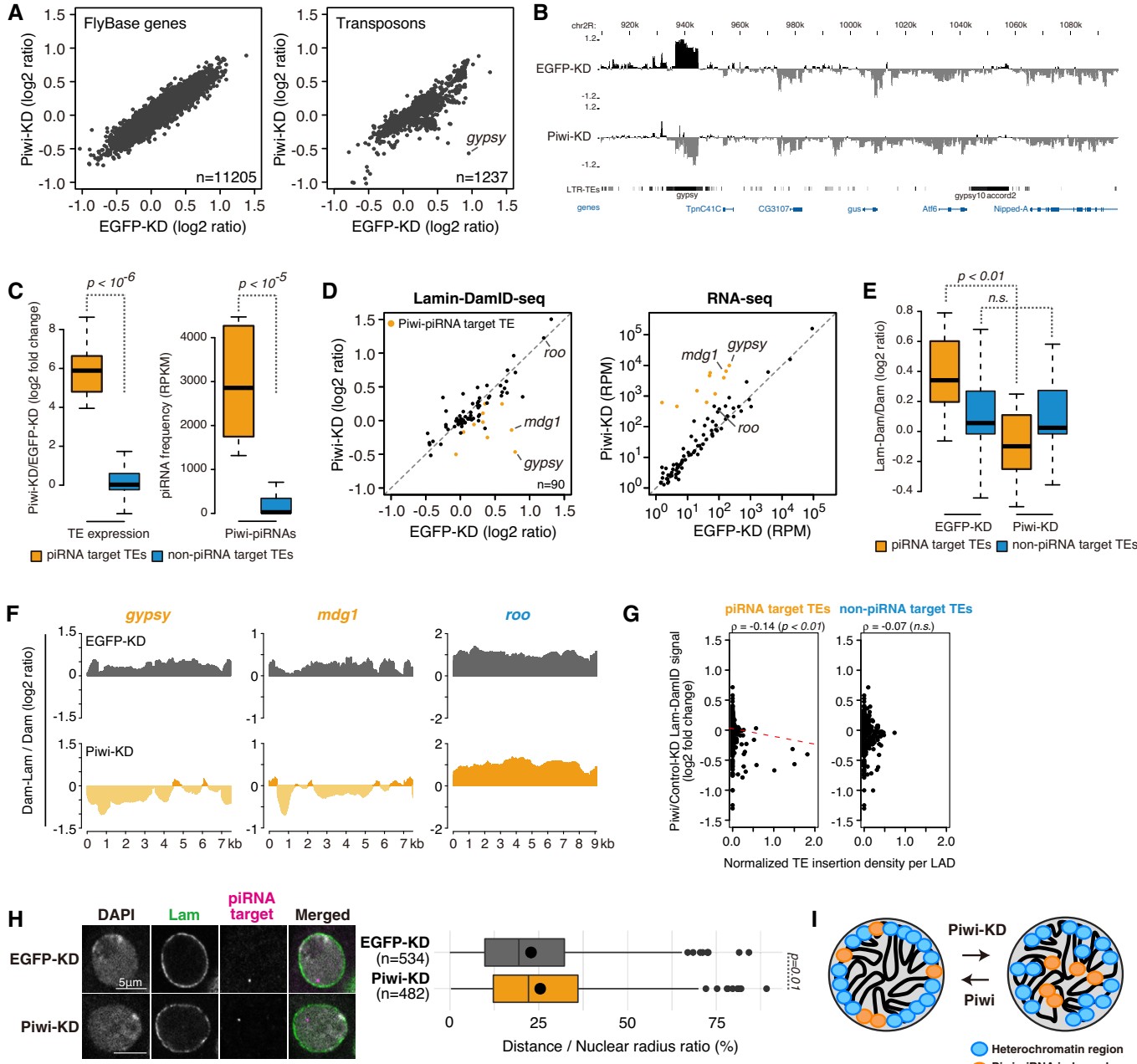

**Figure 2.**

**Figure 2.   Piwi–piRNA target TEs tethering to the nuclear periphery is Piwi-dependent.**

A   Scatterplot of DamID-seq signals (log$_2$ ratio) for FlyBase genes (left) or genome-integrated TEs (right) in EGFP-KD (control, *x*-axis) and Piwi-KD (*y*-axis) samples. Both *x*- and the *y*-axis are a log$_2$ ratio. *Gypsy* signal with the greatest decrease upon Piwi-KD is indicated and shown in genome browser view in (B).

B   DamID-seq signal densities are shown for OSCs with EGFP- and Piwi-KD at genomic regions harboring *gypsy* insertions with the greatest decrease upon Piwi-KD (indicated in (A)). Neighboring regions are annotated with protein-coding genes and LTR TEs. Note that this distribution of the reads includes multi-mapped reads.

C   Boxplots showing fold changes in the expression of piRNA target or non-target TEs based on RNA-seq upon Piwi-KD (left), and the frequency of piRNAs, which target piRNA target TEs or non-piRNA target TEs (right). Definition of piRNA target and non-target TEs is in Fig EV1E. Boxplot whiskers, box, and central band show 1.5 times the inter-quartile range, the first to the third quartile, and median, respectively (*n* = 10 for piRNA target TEs, *n* = 80 for non-piRNA target TEs). *P*-values were calculated using the Wilcoxon rank-sum test.

D   Scatterplot as in (A) for EGFP-KD (control, *x*-axis) or Piwi-KD (*y*-axis) samples. Values were calculated for 90 TE consensus sequences (one value per one type of TE) (left). Scatterplot of RPM values for the same 90 TE consensus sequences in EGFP-KD (control, *x*-axis) or Piwi-KD (*y*-axis) samples examined by RNA-seq (right). The signals for *mdg1*, *gypsy*, and *roo* are labeled within the plots.

E   Boxplot showing DamID-seq signals of piRNA target or non-target TEs upon EGFP (control)- or Piwi-KD. Boxplot whiskers, box, and central band show 1.5 times the inter-quartile range, the first to the third quartile, and median, respectively (*n* = 10 for piRNA target TEs, *n* = 80 for non-piRNA target TEs). *P*-values were calculated using the Wilcoxon rank-sum test.

F   Density plots for DamID-seq signals over consensus sequences from piRNA target (*gypsy* and *mdg1*, orange) and non-target TEs (*roo*, blue) in EGFP (control)- or Piwi-KD cells.

G   Dot plot showing the correlation between the normalized density of indicated TEs per LAD (*x*-axis) and changes in Lam-Dam/Dam signals upon Piwi-KD (*y*-axis). Trend lines are in red. Spearman's rank correlation rho is indicated along with the *P*-value (algorithm AS 89).

H   Oligo-FISH images for the representative piRNA target region, upon EGFP (control)- or Piwi-KD. Gray indicates DAPI staining for DNA, magenta indicates FISH signal, and green indicates Lamin immunofluorescence staining. Scale bar, 5 μm. Ratio of distance from FISH signal to the nuclear periphery (DAPI surface) is quantified at right, and dot in the box plot indicates mean value. Boxplot whiskers, box, and central band show 1.5 times the inter-quartile range, the first to the third quartile, and median, respectively. *P*-values calculated with Mann–Whitney *U*-test.

I   Diagram illustrating relocation of piRNA target TE coding chromosome regions away from the nuclear periphery.

likely affects chromatin conformation (Zheng *et al*, 2018; Ulianov *et al*, 2019). Piwi–piRNA regulation is also coupled to changes in nuclear localization (Fig 2). We therefore asked whether Piwi–piRNA pathways impact chromatin conformation. To analyze the impact of Piwi–piRNA silencing on chromatin conformation, we performed high-throughput chromosome conformation capture (Hi-C) using OSCs depleted of EGFP (control) or Piwi and constructed a Hi-C contact map at a resolution of 40k (Figs 3A and EV2A). When genome-wide contact counts per genomic distance were calculated for Piwi- and Control-KD OSCs, there was only a limited increase in long-range contact counts upon Piwi-KD (Fig EV2B). However, when the difference between Piwi-KD contacts and Control-KD contacts was compared in a locus-specific manner using the contact map, a genome-wide increase in long-range interactions contrasted with a strong decrease in short-range interactions at specific loci (Fig 3B). Relative to the Hi-C dataset of Control-KD OSCs, topologically associating domains (TADs) were defined using HiCExplorer (Ramirez *et al*, 2018). We overlaid the piRNA target TE insertion positions on the differential Hi-C heatmap. This analysis revealed a decrease in short-range interactions within piRNA target TE inserted regions (Fig 3C). Moreover, decreased interactions occurred mostly in intra-TAD interactions. To analyze whether this decrease of intra-TAD interactions at piRNA target TE insertions can be observed genome-wide, normalized intra-TAD interactions upon Piwi-KD were plotted as a heatmap centering piRNA target TE insertions (Fig 3D). This confirmed a decrease in short-range interactions at TE insertions and their neighboring regions. This decrease was not observed when the same analysis was performed using non-piRNA target TEs as controls. We next plotted the number of TE insertions per TAD and normalized changes in intra-TAD interactions per TAD. TADs with higher densities of piRNA target TEs tend to have decreased intra-TAD interactions upon Piwi-KD (Fig 3E). In contrast, TADs with higher densities of piRNA target TEs tend to have increased inter-TAD interactions upon Piwi-KD (Fig 3F). We

further confirmed the decrease of intra-TAD interaction upon Piwi-KD, using oligo-FISH analysis. Two probes were designed to target different regions within the same TAD, and the distance between two signals was calculated to detect the changes in intra-TAD interactions (Fig EV2C). Consistently with the Hi-C data, we were able to observe significant decrease of the distance between two oligo-FISH signals within the same TAD (Fig 3G). These results together suggest a model in which contacts within TADs harboring piRNA target TEs decrease upon Piwi-KD. This loosening of TADs likely results in an increased interactions between different TADs (Fig 3H).

While most TAD boundaries and sizes remained unchanged upon Piwi-KD (Figs EV2D and EV2E), a small population of TAD boundaries identified using Piwi-KD Hi-C data differ from that under Control-KD conditions (Fig 3I). We calculated the TAD separation score for the regions surrounding piRNA target TE insertions and observed a slight decrease in TAD separation score upon Piwi-KD that was specific to piRNA target TE insertions (Fig EV2F and G). Some of these decreases may be creating novel TAD boundaries. 187 out of 225 TAD boundaries were found in both Control- and Piwi-KD cells, and 41 and 38 boundaries were specific to Control- and Piwi-KD cells, respectively (Fig 3J). To further characterize these Control- or Piwi-KD specific boundaries, we focused on piRNA target insertions located at boundary regions. 30% of the piRNA target TE insertions at boundaries were found at Piwi-KD specific boundary regions, whereas the ratio was only 14.1% for non-piRNA target TE insertions (Fig EV2H). In contrast, piRNA target TE insertions at Control-KD specific boundaries were as low as 2.5%, compared to 16.6% for non-piRNA target TE insertions. Based on the decrease in intra-TAD interactions and separation scores, we suggest that piRNA target TE inserted regions can serve as novel TAD boundaries. Since the total number of boundaries is similar between Piwi- and Control-KD (Fig 3J), this is likely due to modulation of boundaries, rather than the formation of entirely new

boundaries. However, when all of the piRNA target TE insertions (including non-boundary insertions) were analyzed, 88% of the piRNA target TE insertions and 92.7% of non-piRNA target TE insertions were found at regions other than boundaries, and TE insertions located at boundary regions were minor (Fig EV2I). This suggests that the majority of piRNA target TE insertions at intra-TAD regions do not result in novel boundary formation, but in some rare cases, possibly dependent on the number of neighboring TE insertions or distance to the nearest boundary, insertions are capable of modulating boundary regions.

It has been previously shown that TADs are associated with LADs (Paulsen *et al*, 2017; Borsos *et al*, 2019), and in *Drosophila*, LAD modulation caused by Lamin-KD leads to changes in TAD contacts (Ulianov *et al*, 2019). We examined whether a similar relationship for Piwi–piRNA-mediated changes of LADs and TADs can be observed. Using LADs annotated by Lamin DamID-seq analysis (Fig 2), we calculated coverage of LADs within each of the defined TADs. The calculated coverage was compared between Piwi- and Control-KD and overlaid with the changes in intra-TAD density (Fig 3K). This revealed that the decrease of LAD coverage per TAD

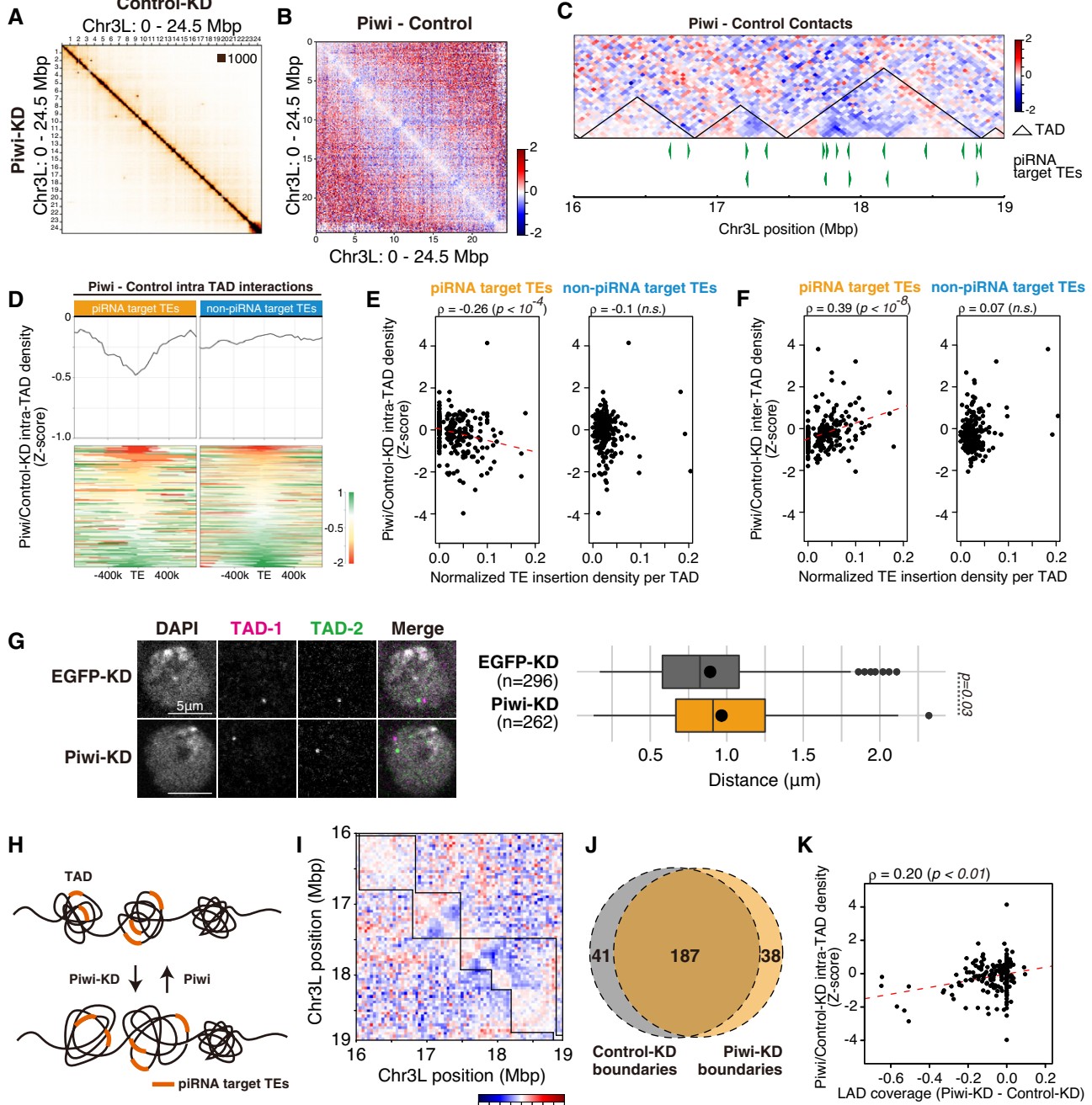

**Figure 3.**

**Figure 3. Piwi–depletion causes a decrease in intra-TAD interactions at piRNA target TE insertion sites.**

A   Hi-C interaction matrices of the Control (EGFP)-KD and Piwi-KD OSCs at Chromosome 3L (40 k resolution).
B   Differential interaction heatmap for chromosome 3L (40k resolution), showing increased (red) and decreased (blue) interacting bins in $\log_2$ fold change upon Piwi-KD.
C   Differential interaction heatmap for 16–19 Mbp of chr3L shown in $\log_2$ fold change. piRNA target TE insertions are indicated at the bottom (green filled triangles), along with TADs defined using Hi-C interactions of Control (EGFP)-KD OSCs (black triangles).
D   Differential intra-TAD interaction density score upon Piwi-KD (upper), centering TE insertions. Heatmap showing changes in the intra-TAD interaction score of each piRNA target TE insertion sites (lower). The results are shown for piRNA target TE and non-piRNA target TE insertions.
E   Dot plot showing the correlation between the normalized density of indicated TEs per TAD (x-axis) and changes in intra-TAD interactions upon Piwi-KD (y-axis). Trend lines are in red. Spearman's rank correlation rho is indicated along with the P-value (algorithm AS 89).
F   Dot plot showing the correlation between the normalized density of indicated TEs per TAD (x-axis) and changes in inter-TAD interactions upon Piwi-KD (y-axis). Trend lines are in red. Spearman's rank correlation rho is indicated along with the P-value (algorithm AS 89).
G   Oligo-FISH images for changes in the intra-TAD interaction at the TAD described in (C), upon EGFP (control)- or Piwi-KD. Gray indicates DAPI staining for DNA, and green and magenta indicate FISH signal at the different positions of TADs (TAD-1 and TAD-2). Details of TAD probe design are described in Fig EV2C. Scale bar, 5 μm. Distance between two FISH signals is quantified at right, and dot in the box plot indicates mean value. Boxplot whiskers, box, and central band show 1.5 times the inter-quartile range, the first to the third quartile, and median, respectively. P-values calculated with Mann–Whitney U-test.
H   Diagram illustrating the possible loss of the intra-TAD interaction for TADs harboring piRNA target TEs.
I   Differential interaction heatmap for 16–19 Mbp of chr3L shown in $\log_2$ fold change. TADs defined using Hi-C interactions of EGFP-KD (control, upper right) or Piwi-KD (lower left) OSCs are overlaid on the heatmap.
J   Venn diagram displaying the number of TAD boundaries detected using EGFP- or Piwi-KD Hi-C interactions.
K   Dot plot showing the correlation between the change in coverage of LADs for each TAD (x-axis), and changes in intra-TAD interactions upon Piwi-KD (y-axis). Trend lines are in red. Spearman's rank correlation rho is indicated along with the P-value (algorithm AS 89).

upon Piwi-KD correlates with a decrease of intra-TAD interaction. These results suggest that Piwi is capable of modulating both chromatin localization and chromatin conformation during regulation of their target TEs.

**Stepwise changes in histone modifications are linked to modulation of nuclear localization and chromatin conformation**

Piwi–piRNA regulation has been shown to correlate with alterations in various histone modifications, including repressive H3K9me3 marks and H1 on target chromatin regions (Sienski *et al*, 2012; Le Thomas *et al*, 2013; Rozhkov *et al*, 2013; Sienski *et al*, 2015; Iwasaki *et al*, 2016). To further characterize the chromatin state, we obtained ChIP-seq data for H3K27Ac and H3K4me3 active histone marks and overlaid the data with Hi-C and Lamin DamID-seq data (Fig 4A). As shown previously, *gypsy* insertion at chromosome 2L is associated with decreased histone H3K9me3 and H1 levels upon Piwi-KD (Sienski *et al*, 2012; Iwasaki *et al*, 2016). The same chromatin region also showed increases in H3K27Ac and H3K4me3, which are associated with active transcription, and decreases of both Hi-C contacts and Lamin DamID-seq signals upon Piwi-KD.

We further analyzed whether changes in histone marks could be globally observed for piRNA target TEs (Fig 4B). Upon grouping of TEs as "piRNA target" and "non-piRNA target" (Figs 2C and EV1E), we observed significant decreases in H3K9me3 and H1 associations and increases in H3K27Ac and H3K4me3 for piRNA-targeted transposons. By looking at ChIP-seq read distributions for representative transposons, we found that decreased H3K9me3 and H1 decreased uniformly across transposon coding regions, whereas H3K27Ac and H3K4me3 increased in LTRs or regions close to LTRs (Fig EV3A). The observed changes in histone marks were found in regions neighboring TE insertions (Figs 4A and EV3B). Consistent with distributions observed within the transposons (Fig EV3A), ChIP-seq signals found in neighboring regions were broader for H3K9me3 and H1 marks and narrower for H3k27Ac and H3K4me3 modifications. Importantly, we could observe that the changes in Hi-C contacts and DamID-seq signals also spread to the neighboring regions of TE insertion (Fig 4A).

We next examined whether these changes in chromatin state at piRNA target TEs are linked to LADs and TADs by comparing changes in LADs and TADs upon Piwi-KD and fold changes in ChIP-seq reads per LAD or TAD (Fig 4C and D). We first analyzed the correlation between histone marks and LAD changes upon Piwi-KD. For each LAD, we plotted the difference between Control- and Piwi-KD for Lamin DamID-seq signals and ChIP-seq reads (Fig 4C). H3K27Ac and H3K4me3 marks significantly increased upon a decrease of Lamin DamID signals. This indicates that LADs whose Lamin contact decreased upon Piwi-KD showed an increase in histone marks associated with active transcription. In contrast, H1 signals were decreased for LADs with decreased Lamin DamID signals. These results were consistent with changes observed for each of the transposons, though a significant correlation could not be observed for H3K9me3 histone marks when analyzed in the LAD context.

Next, we analyzed defined TADs and changes in intra-TAD interactions versus ChIP-seq data (Fig 4D). Interestingly, in the case of TADs, a decrease in repressive H3K9me3 marks and H1 was significantly correlated with decreased intra-TAD interactions, while a significant correlation could not be observed for active histone marks. Also, the correlation between H1 signal fold change and intra-TAD density was higher ($\rho = 0.38$) than that of LAD signal fold change ($\rho = 0.15$). These results indicate that although various histone mark changes were observed within TE insertion sites, active histone marks tend to be associated with changes in LADs, whereas repressive histone marks are instead associated with changes at TADs.

To further dissect changes in histone marks, we used a tethering system applied previously to reveal stepwise regulation by Piwi–piRNAs (Murano *et al*, 2019). Tethering is induced by using the λN-boxB tethering system, in which λN fusion proteins are delivered to a reporter RNA via protein–RNA interaction (Baron-Benhamou *et al*, 2004). Our tethering system uses OSCs carrying a genome-integrated luciferase (*luc*) reporter driven by the ubiquitin promoter, which harbors 14 boxB sites in its intron (Fig EV3C, Murano *et al*, 2019). Expression of λN-HA-tagged piRNA factor led to reporter silencing, coupled with increased H3K9me3 and H1 marks, thus

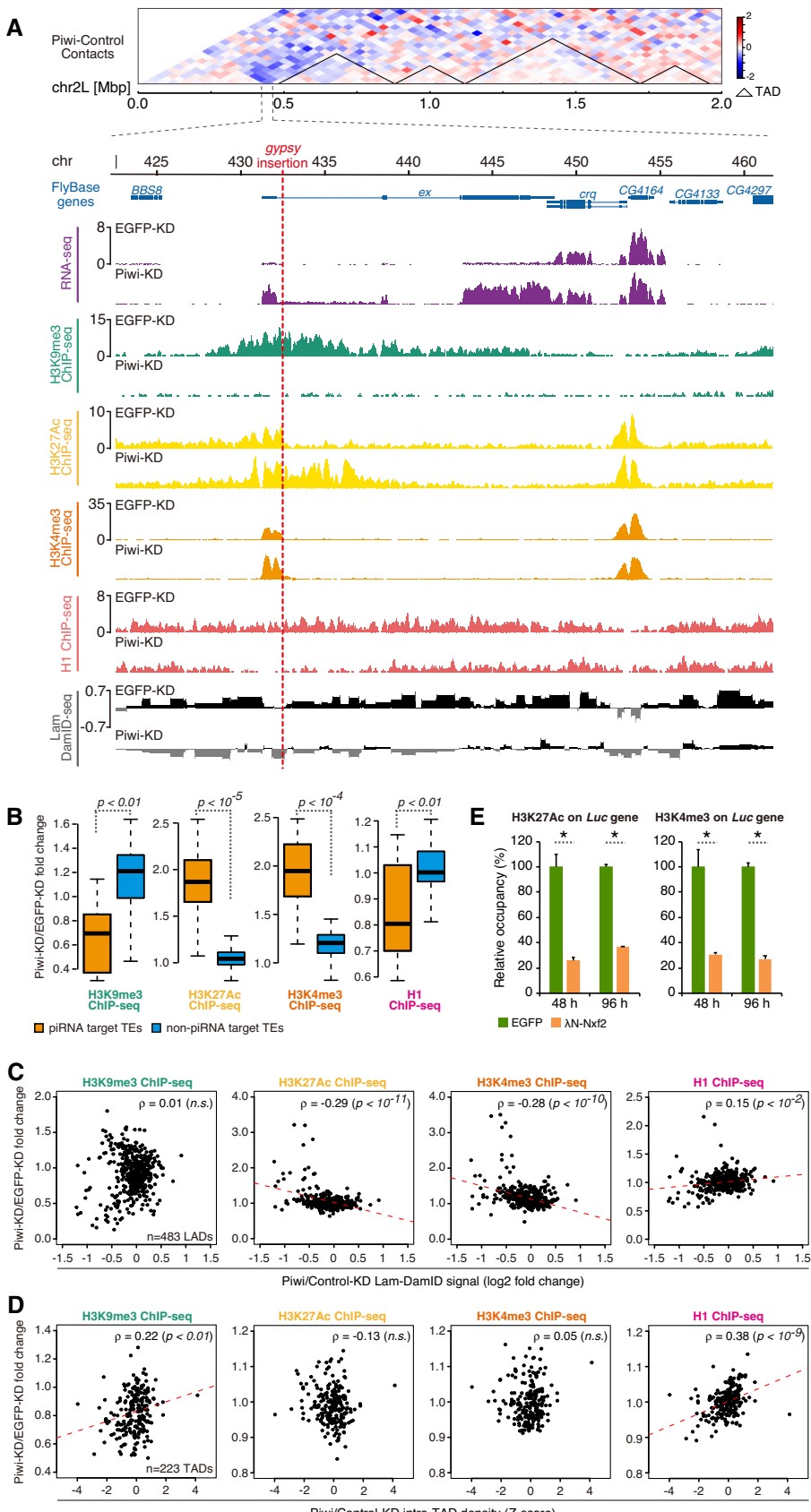

**Figure 4.**

**Figure 4.  Chromatin state is associated with changes in LADs and TADs.**

A  Differential interaction heatmap for Piwi- and Control (EGFP)-KD shown in log$_2$ fold change. Hi-C data is shown with RNA levels (RNA-seq signal density), H3K9me3 association (H3K9me3 ChIP-seq signal density), H3K27Ac association (H3K27Ac ChIP-seq signal density), H3K4me3 association (H3K4me3 ChIP-seq signal density), H1 association (H1 ChIP-seq signal density), and lamin association (Lamin DamID-seq signal density). The genomic region flanking the gypsy insertion (red dashed line) is shown for OSCs with EGFP- and Piwi-KD. FlyBase protein-coding genes are indicated, together with chromosome position.

B  Boxplots showing fold changes in the histone mark association of piRNA target or non-target TEs based on ChIP-seq upon Piwi-KD (definition of piRNA target and non-target TEs is described in Fig EV1E). Boxplot whiskers, box, and central band show 1.5 times the inter-quartile range, the first to the third quartile, and median, respectively (n = 10 for piRNA target TEs, n = 80 for non-piRNA target TEs). *P*-values were calculated by Wilcoxon rank-sum test.

C  Dot plot showing the correlation between changes in Lamin DamID signal upon Piwi-KD (*x*-axis) and changes of indicated ChIP-seq signal in each LADs upon Piwi-KD (*y*-axis). The trend lines are in red. Spearman's rank correlation rho is indicated along with the *P*-value (algorithm AS 89).

D  Dot plot showing the correlation between changes in intra-TAD interactions upon Piwi-KD (*x*-axis) and changes of indicated ChIP-seq signal in each TADs upon Piwi-KD (*y*-axis). The trend lines are in red. Spearman's rank correlation rho is indicated along with the *P*-value (algorithm AS 89).

E  ChIP-qPCR analysis of H3K27Ac and H3K4me3 occupancy on the reporter gene upon transfection of λN-Nxf2 expression vector or control EGFP expression vector. Bar graph shows the occupancy relative to that of the sample transfected with a control plasmid. Error bars indicate SD (n = 3), *P-value < 0.01 (*t*-test).

mimicking Piwi–piRNA regulation. Interestingly, reporter gene expression is regulated at 48 h post-transfection (hpt) of λN-HA-tag fused proteins, but H3K9me3 and H1 marks start to increase at 96 hpt, indicating that silencing occurs in a multi-step manner (Murano *et al*, 2019). The former step involves a decrease of PolII and transcription, but H3K9me3 and H1 levels start to increase at latter step, and how this prior step is linked to transcriptional regulation remains unclear. We further analyzed changes in H3K27Ac and H3K4me3 levels using this tethering system, revealing that these active histone marks were increased at 48 hpt (Fig 4E) and indicating that this change occurs at the prior step. Together with the correlation of LADs with active histone marks and TADs with repressive histone marks (Fig 4C and D), these results suggest a temporal model where Piwi–piRNA affects nuclear positioning of LADs and active histone marks, and this is followed by regulation of TADs and repressive histone marks.

**Nxf2 induces changes in nuclear localization and chromatin conformation at piRNA target regions**

We next asked how the Piwi–piRNA pathway regulates nuclear architecture, as observed in the Piwi-KD. Four groups, including our group, have recently identified the nuclear RNA export factor variant, Nxf2, as being involved in Piwi–piRNA regulation (Batki *et al*, 2019; Fabry *et al*, 2019; Murano *et al*, 2019; Zhao *et al*, 2019). Nxf2 has a similar domain structure to the bulk mRNA exporter, Nxf1, but does not function in RNA export (Herold *et al*, 2000; Herold *et al*, 2001; Herold *et al*, 2003). Earlier reports showed that Nxf2 association with p15/Nxt1, and p15 association with Nxf2, is essential for piRNA-mediated silencing of target TEs (Batki *et al*, 2019; Fabry *et al*, 2019; Murano *et al*, 2019; Zhao *et al*, 2019). Nxf1-associated p15 binds to nuclear pore proteins to export nuclear mRNAs (Forler *et al*, 2004). This prompted us to hypothesize that the Nxf2–p15 complex may serve as a factor that tethers piRNA target TE coding regions to the nuclear periphery.

To test whether Nxf2 can associate with the nuclear periphery, we performed immunoprecipitation of myc tagged Nxf2 or Lamin, followed by Western blotting (Figs 5A and EV4A). This revealed that Nxf2 and Lamin are weakly associated. This weak interaction may be indirect and require an additional factor, perhaps p15. In order to further analyze the impact of Nxf2 on nuclear localization, we first performed Lamin DamID-seq, under Nxf2-KD conditions (Fig EV4B). Similar to Piwi-KD, we observed a decrease in Lamin

DamID signal at *gypsy* coding regions (Fig 5B) and found that other TEs were de-silenced upon Nxf2-KD (Fig 5C). Grouping of piRNA target and non-target TEs (Figs 2C and EV1E) revealed that the DamID signal was specifically decreased for piRNA target TEs (Fig 5D and E) and also at the LAD level (Fig 5F).

We next examined whether this effect on nuclear localization can also affect chromatin conformation using Hi-C analysis (Fig EV4C and D). Changes in Hi-C contacts in Nxf2-KD compared to Control-KD suggest a specific decrease in short-range interactions at particular loci, as seen upon Piwi-KD (Figs 5G and EV4E). This decrease can be observed genome-wide at piRNA target TE insertions (Fig 5H). TADs harboring piRNA target TEs tend to have decreased intra-TAD interactions and increased inter-TAD interactions (Fig 5I and J), similar to Piwi-KD. A slight decrease in the TAD separation score upon Nxf2-KD at piRNA target TE coding regions was observed (Fig EV4F). However, the number of boundaries specific to Nxf2-KD was 33, and 200 were in common with Control-KD (Fig EV4G), suggesting that TAD borders are not highly affected upon Nxf2-KD. We also observed a correlation between LAD coverage per TAD and changes in intra-TAD density in Nxf2-KD, similar to Piwi-KD (Fig 5K). These results suggest that Nxf2 serves as an essential factor for the regulation of nuclear localization and chromatin conformation upon Piwi–piRNA-mediated regulation. To investigate the time point at which piRNA target regions are localized to the nuclear periphery, we made use of the Nxf2 tethering system (Murano *et al*, 2019; Fig EV3C). We followed the nuclear location of the reporter using oligo-FISH, 48 and 96 h after transfection of the λN-Nxf2 expression vector (Fig 5L). This revealed that the distance between the nuclear periphery and FISH signal significantly decreased after 48 h and was stable by 96 h. Since the decrease in active histone marks occurs 48 h prior to the changes in repressive histone marks (Fig 4E), this was consistent with the result that the changes in LADs are correlated with active histone marks (Fig 4C). These observations suggest that Nxf2 first localizes target TE regions to the nuclear periphery which is coupled to a decrease in active histone marks and PolII. This is followed by an increase in H3K9me3 and H1 and chromatin conformational change.

**Heterochromatin factors cooperatively function to induce chromatin conformational changes**

Since chromatin conformational changes in Piwi–piRNA pathway were linked to repressive chromatin states (Fig 4), we performed Hi-C analysis after KD of different factors involved in late steps of

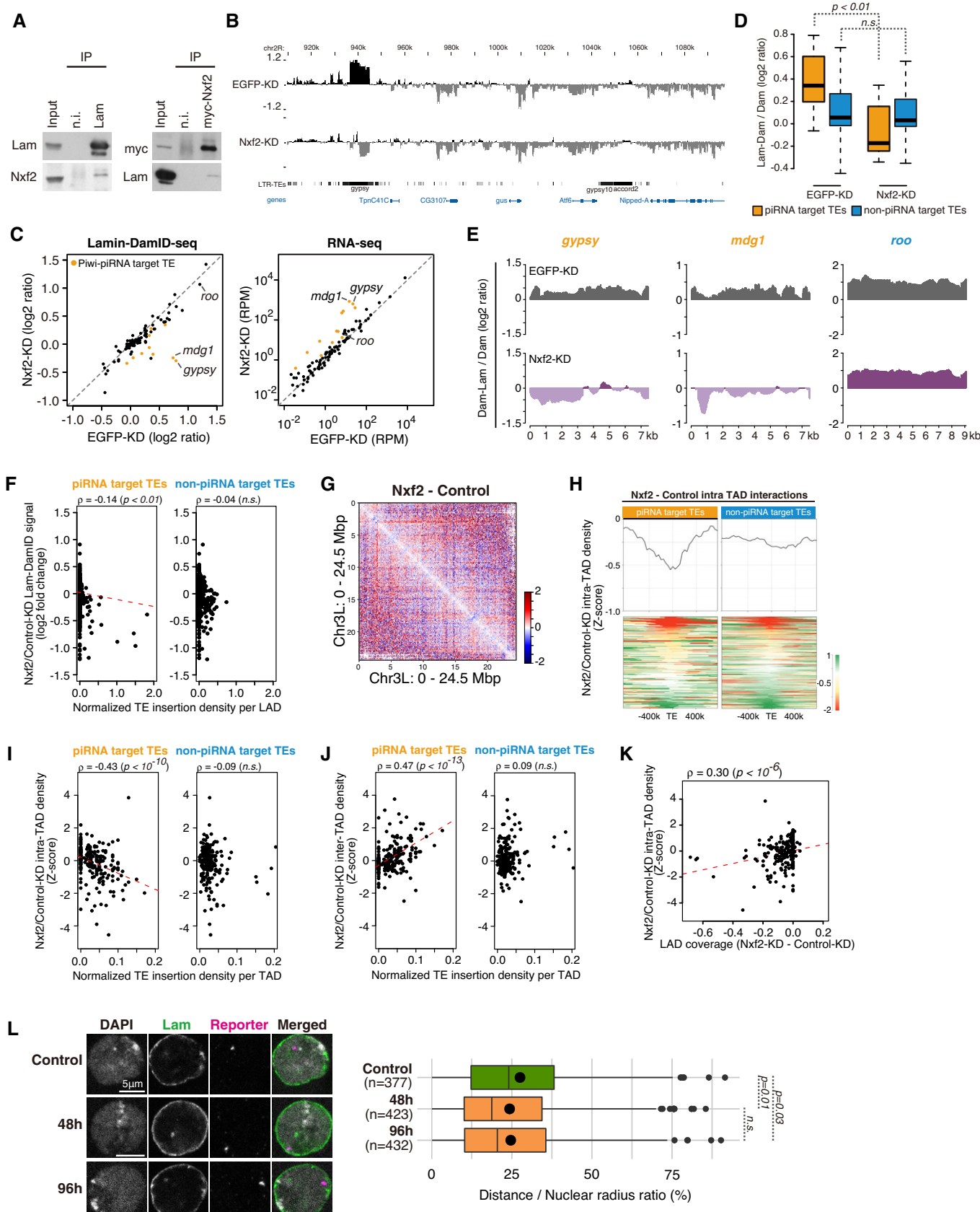

Figure 5.

**Figure 5.  Nxf2 regulates chromatin localization and decreases of intra-TAD interaction.**

A   Immunoprecipitation (IP) from OSC lysate using anti-Lam, or myc antibody, followed by WB using indicated antibodies. Mouse immunoglobulin G (n.i.) was used for control IP. Note that OSCs used at the right panel are a line which stably expresses myc-Nxf2.

B   DamID-seq signal densities are shown for OSCs with EGFP (control)- and Nxf2-KD at genomic regions harboring *gypsy* insertions with the greatest decrease upon Nxf2-KD (locus shown in Fig 2B). Neighboring regions are annotated with protein-coding genes and LTR TEs. Note that this distribution of the reads includes multi-mapped reads.

C   Scatterplot of DamID-seq signals (log₂ ratio) for EGFP (control)-KD (x-axis) or Nxf2-KD (y-axis) samples. Values were calculated for 90 TE consensus sequences (one value per one type of TE) (left). Scatterplot of RPKM values for the same 90 TE consensus sequences in EGFP-KD (control, x-axis) or Piwi-KD (y-axis) samples examined by RNA-seq (right). The signals for *mdg1*, *gypsy*, and *roo* are labeled within the plots.

D   Boxplot showing Lamin DamID-seq signals of piRNA target or non-target TEs upon EGFP (control)- or Nxf2-KD. Boxplot whiskers, box, and central band show 1.5 times the inter-quartile range, the first to the third quartile, and median, respectively (n = 10 for piRNA target TEs, n = 80 for non-piRNA target TEs). P-values were calculated using the Wilcoxon rank-sum test.

E   Density plots for DamID-seq signals over consensus sequences from piRNA target (*gypsy* and *mdg1*, orange) and non-target TEs (*roo*, blue) in EGFP (control)- or Nxf2-KD cells.

F   Dot plot showing the correlation between the normalized density of indicated TEs per LAD (x-axis) and changes in Lam-Dam/Dam signals upon Nxf2-KD (y-axis). Trend lines are in red. Spearman's rank correlation rho is indicated along with the P-value (algorithm AS 89).

G   Differential interaction heatmap for chromosome 3L (40k resolution) in log₂ fold change, showing increased (red) and decreased (blue) interacting bins upon Nxf2-KD.

H   Differential intra-TAD interaction density score upon Nxf2-KD (upper), centering TE insertions. Heatmap showing changes in the intra-TAD interaction score of each piRNA target TE insertion sites (lower). The results are shown for piRNA target TE and non-piRNA target TE insertions.

I   Dot plot showing the correlation between the normalized density of indicated TEs per TAD (x-axis) and changes in intra-TAD interactions upon Nxf2-KD (y-axis). Trend lines are in red. Spearman's rank correlation rho is indicated along with the P-value (algorithm AS 89).

J   Dot plot showing the correlation between the normalized density of indicated TEs per TAD (x-axis) and changes in inter-TAD interactions upon Nxf2-KD (y-axis). Trend lines are in red. Spearman's rank correlation rho is indicated along with the P-value (algorithm AS 89).

K   Dot plot showing the correlation between the change of coverage of LAD for each TAD (x-axis), and changes in intra-TAD interactions upon Nxf2-KD (y-axis). The trend lines are in red. Spearman's rank correlation rho is indicated along with the P-value (algorithm AS 89).

L   Oligo-FISH images for the reporter of the tethering system, at 0 h (control), 48 h, and 96 h after transfection of λN-Nxf2 expression vector. Gray indicates DAPI staining for DNA, magenta indicates FISH signal, and green indicates Lamin staining. Scale bar, 5 μm. The ratio of distance from FISH signal to the nuclear periphery (DAPI surface) is quantified at right, and dot in the box plot indicates mean value. Boxplot whiskers, box, and central band show 1.5 times the inter-quartile range, the first to the third quartile, and median, respectively. P-values calculated with Mann–Whitney U-test.

Source data are available online for this figure.

Piwi–piRNA regulation, namely H1, methyltransferase Egg, and HP1a (Sienski *et al*, 2012; Le Thomas *et al*, 2013; Rozhkov *et al*, 2013; Sienski *et al*, 2015; Yu *et al*, 2015; Iwasaki *et al*, 2016; Fig EV5A). The resulting differential Hi-C heatmap shows that H1 depletion causes similar changes to those seen upon Piwi depletion, inducing a decrease in short-range interactions at specific loci (Fig 6A). In contrast, Egg- and HP1a-KD tend to have a larger impact on the global decrease of short-range interactions (Fig 6A), possibly due to the larger impact on transcriptome upon knockdown of these genes. The increase in short-range interactions was remarkable, especially for HP1a-KD, and is also observed for the global analysis of contact counts per genomic distance (Fig 6B). Since this global increase in short-range interaction is independent of piRNA target insertions, we hypothesized that a general cell state, such as the cell cycle, is affected. To test this, we analyzed the cell cycle of OSCs upon KD of Piwi and HP1a, using propidium iodide staining followed by FACS analysis (Fig EV5B). For HP1a-KD, we found a decrease in populations at the G1 state and increase in populations at the G2-M state, where the impact on cell cycle upon Piwi-KD was limited. In mouse ES cells, it has previously been shown that short-range contacts are lower in G1 state and higher in G2-M state (Nagano *et al*, 2017). Therefore, the global increase in short-range interactions observed upon HP1a-KD may be due to their effect on cell cycle state.

We also defined TADs for these KD samples and found that the decrease in short-range intra-TAD interactions in H1-KD correlated with piRNA target TE insertions (Fig 6C). Also, a slight decrease in intra-TAD interactions was observed for Egg- and HP1a-KD, though not as strong as in the H1-KD. Consistently, when we analyzed the piRNA target TE insertions and decrease in intra-TAD interactions

per TAD, weak correlation was detected for H1-KD, but not for Egg-nor HP1a-KD (Fig 6D). In the case of inter-TAD interactions, the correlation between increased interactions and piRNA target TE insertions per TAD was observed for H1- and Egg-KD (Fig 6E). However, for the Egg- and HP1a-KD, a slight correlation between inter-TAD interactions was also observed for non-piRNA target insertions, suggesting that Egg and HP1a may affect inter-TAD interactions with TE insertions independent of Piwi–piRNA regulations.

For TAD boundaries, a decreased TAD separation score at piRNA target TE insertions could also be observed at these KD samples, though this is not as strong as in the Piwi-KD (Figs EV5C and EV2F). The number of novel TAD boundaries within KD samples was larger than that of Piwi-KD, especially for H1- and HP1a-KD (Fig EV5D). This may be because these factors broadly impact gene expression beyond piRNA target TEs. In line with this, most of these H1- and HP1a-KD specific boundaries do not overlap with Piwi-KD specific TADs that overlap with TEs (Fig EV5D), indicating many novel TAD boundaries found under these conditions are independent of Piwi–piRNA regulation. In contrast, 7 out of 9 novel TAD boundaries observed specifically for the Piwi-KD sample overlapped with novel TAD boundaries seen in Egg-KD samples. This suggests that H3K9me3 modification plays a key role in boundary modulation at piRNA target inserted regions.

These data indicate that downstream factors of the Piwi–piRNA pathway, H1, Egg, and HP1a, have a different impact on chromatin conformational change caused by Piwi–piRNA regulation. Loss of H1 has a higher impact in terms of both increased intra-TAD interactions and decreased inter-TAD interactions at piRNA target TE insertions, and Egg may contribute to boundary formation. We have previously shown that H1 and HP1a function in parallel pathways

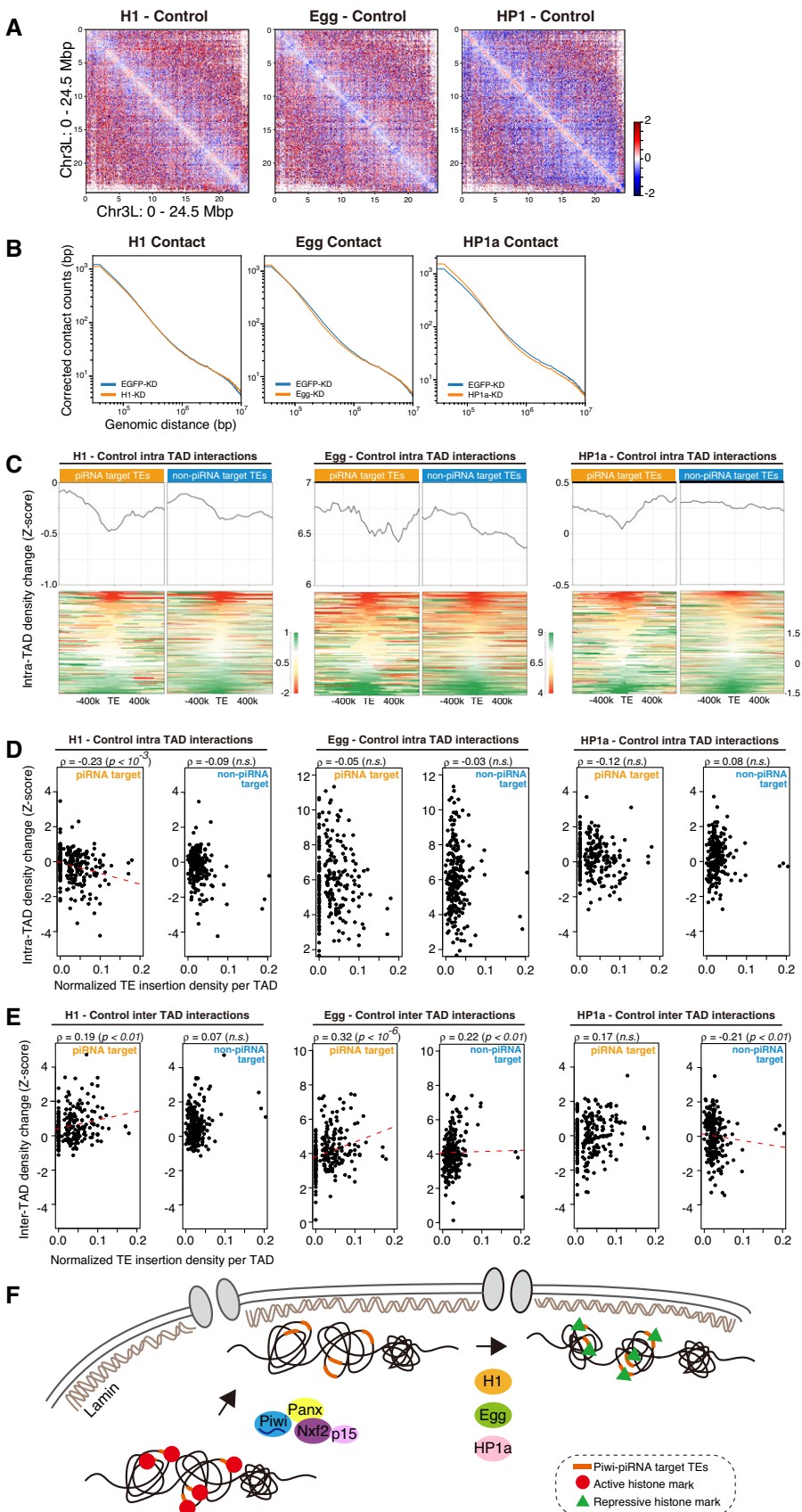

**Figure 6.**

**Figure 6. Downstream factors, H1, Egg, and HP1a, cooperatively regulate chromatin conformation in the Piwi–piRNA pathway.**

A   Differential interaction heatmap for chromosome 3L (40 k resolution) in $\log_2$ fold change, showing increased (red) and decreased (blue) interacting bins upon H1(left)-, Egg(middle)-, or HP1a(right)-KD.

B   Relative contact probability (RCP) plot showing corrected contact counts at different genomic ranges, genome widely. RCP values are compared between EGFP-KD (blue) and H1(left)-, Egg(middle)-, or HP1a(right)-KD OSCs.

C   Differential intra-TAD interaction density score upon H1 (left)-, Egg (middle)-, or HP1a (right)-KD (upper), centering TE insertions. Heatmap showing changes in the intra-TAD interaction score of each piRNA target TE insertion sites (lower). The results are shown for piRNA target TE (left) and non-piRNA target TE (right) insertions.

D   Dot plot showing the correlation between the normalized density of indicated TEs per TAD (x-axis) and changes in intra-TAD interactions upon H1(left)-, Egg (middle)-, or HP1a(right)-KD (y-axis). Trend lines are in red. Spearman's rank correlation rho is indicated along with the *P*-value (algorithm AS 89). n.s.; not significant ($P > 0.05$)

E   Dot plot showing the correlation between the normalized density of indicated TEs per TAD (x-axis) and changes in inter-TAD interactions upon H1(left)-, Egg(middle)-, or HP1a(right)-KD (y-axis). Trend lines are in red. Spearman's rank correlation rho is indicated along with the *P*-value (algorithm AS 89). n.s.; not significant ($P > 0.05$).

F   Schematic model of stepwise Piwi–piRNA-mediated regulation of nuclear architecture and chromatin state.

in a synergistic manner in the Piwi–piRNA pathway (Iwasaki *et al*, 2016). Therefore, H1, Egg, and HP1a, all of which function downstream of Piwi and Nxf2, would cooperate to induce chromatin conformational change at piRNA TE coding loci.

## Discussion

Here, we propose a model where Piwi–piRNA-mediated TE regulation of targets is linked to changes in nuclear localization, chromatin conformation, and histone modifications (Fig 6F). Interestingly, this occurs in a stepwise manner, where localization and removal of active histone marks is followed by repressive histone marks and chromatin conformational change. These results suggest that nuclear small RNA-mediated regulation modifies nuclear architecture at the genomic level.

A general conundrum in this field is whether nuclear positioning controls gene expression or vice versa (Luperchio *et al*, 2014; van Steensel & Furlong, 2019). There is some evidence that transcription can counteract lamina interactions and modulate Hi-C contacts. The relocation of piRNA target TEs after Piwi- or Nxf2-KD could be a secondary effect of transcriptional activation. Meanwhile, slight de-repression of piRNA target TEs upon the depletion of lamin proteins (Fig 1E) may suggest a possible defect in transcription upon disruption of lamina association. Defects in transcriptional regulation and nuclear localization upon Piwi-KD occur prior to changes in H3K9me3 histone marks and chromatin conformation (Figs 4 and 5L) (Murano *et al*, 2019). In *C. elegans*, heterochromatin regions with H3K9me3 histone marks are anchored to perinuclear regions (Towbin *et al*, 2012). This suggests that in the Piwi–piRNA pathway, deposition of H3K9me3 at a later step reinforces the association of chromatin regions with perinuclear regions. Indeed, Piwi or Nxf2 knockdown, which decreases the Lamin DamID-seq signal at Piwi–piRNA target regions (Figs 2 and 5), also results in a major decrease in H3K9me3 levels (Batki *et al*, 2019; Fabry *et al*, 2019; Murano *et al*, 2019; Zhao *et al*, 2019). These results suggest that lamina localization may both be the cause and result of TE repression. Nuclear peripheral localization may affect gene expression to some extent, and the regulation of gene expression may be enforced due to its localization to the nuclear periphery.

We proposed that the Piwi–piRNA pathway triggers spatial regulation of genomic regions in order to repress transposable elements. In other words, Piwi–piRNA system may be able to send target regions to specific positions within the nucleus in order to

efficiently regulate their expression. In OSCs, piRNAs are generated and loaded onto Piwi protein at the cytoplasmic Yb body (Saito *et al*, 2010). Therefore, target gene localization to the nuclear periphery may facilitate regulation by Piwi–piRNAs imported from the cytoplasm. Within the nucleus, Piwi–piRNA functions together with Panx-Nxf2-p15 at the nascent transcript, in order to silence target TEs (Batki *et al*, 2019; Fabry *et al*, 2019; Murano *et al*, 2019; Zhao *et al*, 2019). Recent studies have suggested that Panx-Nxf2-p15 complex forms condensates upon Piwi–piRNA-mediated silencing (Eastwood *et al*, 2021; Schnabl *et al*, 2021). Condensate formation is coupled to dimerization of Panx-Nxf2-p15 complex, and piRNA target regions may therefore be spatially regulated upon silencing. Moreover, it has been proposed that condensate formation is stimulated by nascent RNA produced during early steps in transcription initiation (Henninger *et al*, 2021). By analogy with these observations, nuclear localization and chromatin conformational changes observed in this study may also be linked to condensate formation and possibly form a spatial structure for Piwi–piRNA regulation.

Although Piwi–piRNA regulation affects nuclear localization of target chromatin regions, the effect on TAD boundaries was limited, and the effect mainly observed as changes in intra- and inter-TAD interactions (Fig 3). In line with this observation, it has been reported that even depletion of nuclear lamins does not result in modulation of TAD boundaries in *Drosophila* or mouse ES cells (Zheng *et al*, 2018; Ulianov *et al*, 2019). This suggests that changes in nuclear localization of target chromatin regions are not directly linked to modulation of TAD boundaries, but Piwi–piRNA regulation affects expression levels of genes that are localized within the same TAD as piRNA target TEs. This is consistent with the fact that Piwi–piRNA regulation regulates not only expression, histone modifications, and chromatin accessibility of its target TEs, but also their neighboring regions (Sienski *et al*, 2012; Iwasaki *et al*, 2016).

Recent studies have revealed that heterochromatin structure is dynamic and relatively large molecules can move between both heterochromatin and euchromatin regions (Hihara *et al*, 2012). In addition, some essential genes have been reported to be encoded and transcribed in heterochromatin (Marsano *et al*, 2019). The Piwi–piRNA system, which is reversible and can be reversed upon loss of Piwi proteins or specific piRNAs, may contribute to this dynamic regulation of heterochromatic regions. Since Piwi proteins and piRNAs are specifically expressed in the germline, it may be the case that the Piwi–piRNA system has come to regulate nuclear architecture in these inheritable populations. It is tempting to

speculate that gene expression and nuclear localization of chromatin is designed to account for TE regulation by Piwi–piRNAs. Inversely, TEs may even function as "functional motifs" in order to rewire nuclear architecture (Ohtani & Iwasaki, 2021). The dynamics of heterochromatin regulation by the Piwi–piRNA pathway therefore provides insight into complex gene regulation controlled by TEs and even impacts the genome at the level of nuclear architecture.

# Materials and Methods

### Cell lines

Ovarian somatic cells (OSCs) were established in previous studies (Niki *et al*, 2006; Saito *et al*, 2009). OSCs were cultured at 26°C in Shield and Sang M3 Insect Medium (Caisson Laboratories) supplemented with 10% fly extract, 10% fetal bovine serum, 0.6 mg/ml glutathione, and 10 μg/ml insulin.

### Cell transfection

For siRNA transfection into OSCs, 200 pmol siRNA duplex was nucleofected into $3.0 \times 10^6$ cells using the Cell Line 96-well Nucleofector Kit SF (Lonza) and program DG150 of the 96-well Shuttle Device (Lonza) or Cell Line Nucleofector Kit V (Lonza) and program T-029 of the Nucleofector II Device (Lonza). The siRNAs were transfected twice for 4-day KD and once for 2-day KD. Piwi-KD was performed for 4 days, while Lam/LamC-KD was performed for 2 days due to the severe damage caused to cells. siRNA sequences are listed in Table EV2. All of the siRNAs were tested for their efficiency to silence target genes in the previous study (Iwasaki *et al*, 2016; Murano *et al*, 2019). Co-transfection of siRNA and plasmid DNA was performed using the Cell Line Nucleofector Kit V (Lonza) and program T-029 of the Nucleofector II Device (Lonza).

### Immunoprecipitation

To obtain whole-cell lysate, OSCs were washed in PBS and lysed in IP buffer (50 mM HEPES-KOH pH 7.4, 200 mM KCl, 1 mM EDTA, 1% Triton X-100, 0.1% Na-deoxycholate or 30 mM HEPES-KOH pH 7.4, 150 mM KOAc, 5 mM MgOAc, 5 mM DTT, 0.1% NP40) followed by syringe homogenization using 30G needle and probe sonication (Bioruptor). After centrifugation, the supernatant was used for immunoprecipitation. Antibodies were immunized on Dynabeads Protein G (Thermo Fisher) and incubated with lysates for 2–4 h, at 4°C. Then, beads were washed three to five times in IP buffer. Anti-Piwi (Saito *et al*, 2006), anti-Lam (DHSB, ADL84-12), and anti-LamC (DSHB, LC28.26) antibodies were used, and control immunoprecipitation using mouse nonimmune IgG (IBL) was conducted in parallel.

### Western blotting

Western blotting was performed as described previously (Saito *et al*, 2006). Anti-Piwi (Saito *et al*, 2006), anti-Lam (DHSB, ADL84-12), anti-LamC (DSHB, LC28.26), and anti-Tubulin (DSHB, E7) antibodies were used. Intensity of the bands was quantified using ImageJ software (Schneider *et al*, 2012).

### Immunofluorescence

Immunofluorescence of OSCs was performed using anti-Lam (DHSB, ADL84-12) or anti-LamC (DSHB, LC28.26) antibody as described previously (Iwasaki *et al*, 2016). Indicated antibodies were used for a primary antibody, and Alexa Fluor488-conjugated anti-mouse IgG1 (Molecular Probes) (1:1,000 dilution) was used as a secondary antibody. Slides were mounted using VECTASHIELD Mounting Medium with DAPI (Vector Laboratories).

### Quantitative reverse-transcription PCR

qRT–PCR analysis was performed as previously described (Iwasaki *et al*, 2016). Oligonucleotides used for qRT–PCR primers are described in Table EV2.

### Tethering assay using OSCs

Tethering assay was performed as described previously using the same system (Murano *et al*, 2019). A stable line of OSC with reporter construct of 14× boxB at the intron of luciferase (pUb-14boxB-Fluc) was used together with λN-HA fusion protein expression constructs that were adopted from our previous study (Murano *et al*, 2019). For the tethering assay, $3 \times 10^6$ OSCs were transfected with 5 μg of plasmid DNA using the Cell Line Nucleofector Kit V (Lonza) and Program T-029 of the Nucleofector II Device (Lonza) and passaged in four wells of a 24-well tissue culture plate. Cells were harvested for ChIP-qPCR assay after 48 h or 96 h.

### Chromatin enrichment for proteomics

Chromatin enrichment for proteomics (ChEP) was performed as described previously (Kustatscher *et al*, 2014). Briefly, OSCs were fixed with formaldehyde, lysed, and digested with RNase. Nuclei was resuspended with 4% SDS buffer and mixed with 8 M urea buffer. Nuclei were washed with SDS buffer and storage buffer followed by ultrasonication for DNA shearing and boiling the samples to reverse formaldehyde cross-links. Samples were reduced, alkylated, and digested by phase transfer surfactant-aided digestion method (Masuda *et al*, 2008). Peptides were desalted on an C18-SCX StageTips (Adachi *et al*, 2016) and applied LC-MS/MS (coupling an UltiMate 3000 Nano LC system (Thermo Scientific, Bremen, Germany) and an HTC-PAL autosampler (CTC Analytics, Zwingen, Switzerland) to a Orbitrap Elite mass spectrometer (Thermo Scientific)). Peptides were delivered to an analytical column (75 μm × 30 cm, packed in-house with ReproSil-Pur C18-AQ, 1.9 μm resin, Dr. Maisch, Ammerbuch, Germany) and separated at a flow rate of 280 nl/min using a 145-min gradient from 5% to 30% of solvent B (solvent A, 0.1% FA and 2% acetonitrile; solvent B, 0.1% FA and 90% acetonitrile). The instrument was operated in the data-dependent mode. Survey full scan MS spectra (m/z 300–1,500) were acquired in the Orbitrap with 30,000 resolution after accumulation of ions to a $1 \times 10^6$ target value. Dynamic exclusion was set to 60 s. The 20 most intense multiplied charged ions ($z \geq 2$) were sequentially accumulated to a $1 \times 10^5$ target value and fragmented in the collision cell by CID with a maximum injection time of 50 ms. Typical mass spectrometric conditions were as follows: spray voltage, 2 kV; heated capillary temperature, 200°C;

and normalized HCD collision energy, 35%. The MS/MS ion selection threshold was set to $1 \times 10^3$ counts. A 2.0 Da isolation width was chosen. Raw MS data were processed by MaxQuant (version 15.1.2) supported by the Andromeda search engine. The MS/MS spectra were searched against the UniProt *Drosophila* melano database with the following search parameters: full tryptic specificity, up to two missed cleavage sites, carbamidomethylation of cysteine residues set as a fixed modification, and N-terminal protein acetylation and methionine oxidation as variable modifications. The false discovery rate of protein groups, peptides, was less than 0.01.

## Proteomics data analysis

Total of six replicates (three biological replicate samples and two technical replicates for each sample) were obtained per sample. Fold change for Piwi-KD sample and EGFP-KD sample was calculated for each experiment, and *t*-test was performed in order to calculate *P*-value of obtained fold changes. Adjusted *P*-value (Bonferroni test) was calculated for replicate samples, and proteins with adjusted *P*-value under 0.05 were shown in red. Volcano plot was created using R. Maximum $\log_{10}$ *P*-value was set to 6. Argk protein, whose expression level was decreased upon Piwi-KD, was the only protein which was out of this range ($\log_{10}$ *P*-value of 7.2, $\log_2$ FC of −2.17). 25 proteins with adjusted *P*-value under 0.05 were listed and analyzed for enriched Gene Ontology terms using DAVID v6.8 (Jiao *et al*, 2012). Terms with *P*-value under 0.1 were extracted.

## mRNA-seq and small RNA-seq data analysis

Small RNA-seq data and mRNA-seq data upon Piwi- or Nxf2-KD were obtained from the previous studies (Iwasaki *et al*, 2016; Data ref: Iwasaki *et al*, 2016; Murano *et al*, 2019; Data ref: Murano *et al*, 2019). Bowtie 2.2.9 (Langmead & Salzberg, 2012) was used with the default parameters to map sequences to the *Drosophila genome* (dm3). The positions of TE insertions within the genome were adopted form the previous study (Sienski *et al*, 2012; Data ref: Sienski *et al*, 2012). Reads mapped to the genome were then mapped to the *Drosophila melanogaster* TE consensus sequence from Repbase (Jurka, 1998), using Bowtie 1.0.1, and those that were uniquely mapped were extracted. Expression levels (RPM) of TEs were calculated using reads that were mapped to the TE consensus sequence. Expression levels (RPKM) of FlyBase genes were calculated using TopHat 2.0.14 (Trapnell *et al*, 2009) and DESeq2 (Love *et al*, 2014). *Drosophila melanogaster* (dm3) GTF files obtained from Illumina iGenomes were used to define known gene structures.

## Hi-C analysis

Hi-C using OSCs were performed as described previously (Ulianov *et al*, 2016) with some minor modifications. OSCs were fixed with 1% formaldehyde for 10min and stored at −80° before usage. $5 \times 10^5$ cells were lysed and chromatin digested using 30G needle. Chromatin digestion was performed using HindIII restriction enzyme. Ends were filled with biotin-14-dCTP followed by blunt-end ligation. Proteins were reverse-crosslinked from the cells, and DNA purification was performed. Biotin was removed from un-ligated ends, and DNA was sonicated using covaris S220 (covaris). Sonicated DNA was used for biotin pull down by streptavidin conjugated to Dynabeads® MyOne™ Streptavidin C1 beads (Invitrogen). Pulled-down DNA was used as the template to construct DNA library using NEBNext Ultra II DNA Library Prep Kit for Illumina (NEB). The prepared library was checked for its quality by using NheI digestion. Obtained libraries were sequenced using HiSeq4000 (Illumina) or HiSeq X (Illumina) instrument.

## Hi-C data analysis

We first identified the highest appropriate resolution for our Hi-C datasets by analyzing the reproducibility between replicates (Yardimci *et al*, 2019). Specifically, we visualized the reproducibility scores produced by GenomeDISCO (Ursu *et al*, 2018), HiCRep (Yang *et al*, 2017), and QuASAR-Rep (Yardimci *et al*, 2019) software to analyze efficient resolution that can be used to perform Hi-C analyses, and used 40 kb resolution, where the scores begin to plateau, for all subsequent analyses. HiCExplorer v3.3 software (Ramirez *et al*, 2018) was basically used for the calculation of Hi-C matrix, and GENOVA v1.0 was used for visualizing the results. Two replicates were performed per knockdown, and hicCorrelate was used to calculate correlation between replicate samples. Due to the high correlation between replicates, samples were combined by using hicSumMatrices for the further analysis. Bins of too low or high coverage were removed, and the Hi-C matrix was iteratively corrected by ICE using hicCorrectMatrix. Hi-C matrix was further normalized between different knockdown samples (e.g., EGFP vs Piwi) using hicNormalize. Hi-C differential heatmaps were created using hicCompareMatrices, and relative contact probability per genomic distance plots was created using hicPlotDistVsCounts. TADs were defined using hicFindTADs, and matrixes were converted into ginteractions file using hicConvertFormat. Hi-C interaction counts involving 40 kb genomic bins located in the same TAD were summed together to create intra- and inter-TAD Hi-C matrices. To evaluate the enrichment of intra- and inter-TAD interactions involving specific TAD(s), the background distributions of intra- and inter-TAD interactions were estimated by randomly sampling genomic regions with the same sizes as TADs from the same chromosomes and calculating the means and standard deviations of observed interaction counts. This scheme essentially reduced biases in Hi-C interaction counts due to TAD sizes and chromosomes. Then, the Z-score for each intra- and inter-TAD interaction was calculated using the formula Z-score = ($\log_2$ fold change − average $\log_2$ fold change)/standard deviation of $\log_2$ fold change. Lastly, ginteractions file was converted to hicpro format using in-house script and used as an input for the visualization using GENOVA. Hi-C contact heatmaps and insulation heatmaps were created using GENOVA.

## DamID-seq analysis

pNDamMyc plasmid (van Steensel & Henikoff, 2000) was a kind gift from Dr. Bas van Steensel. Lamin was cloned into pNDamMyc vector as described previously (Pickersgill *et al*, 2006). pNDamMyc plasmid was digested using NotI and XbaI, and amplified Lamin sequence was cloned into the plasmid by using In-Fusion HD cloning system (Clontech). Cloned sequence was confirmed by sequencing. Oligonucleotides used for amplification of Lamin are described in Table EV2. DamID-seq was performed as described

previously with some modifications (Pickersgill *et al*, 2006). DamID vectors were transfected into OSCs using Nucleofector as described in the "Cell transfection" section, one day before cell harvest. Genome DNA precipitation, DpnI digestion, adapter ligation, DpnII digestion, PCR, and PCR cleanup were performed as described within "Van Steensel Lab version 28/08/2006" protocol. Obtained DNA was used as the template to prepare deep sequencing library by using NEB Next Ultra II DNA library prep kit (NEB). Libraries were sequenced by using HiSeq X Ten instrument (Illumina).

### Dam-ID-seq data analysis

Adapters were removed using Cutadapt, and reads shorter than 50nt after adapter removal were discarded. damidseq_pipeline (Marshall & Brand, 2015) was used to map the reads to *Drosophila melanogaster* (dm3) genome, followed by normalization and background reduction. Within damidseq_pipeline, bowtie2 is used for mapping the reads with default parameter, which also permits multi-mapped reads. The multi-mapped reads are mapped to only one of the best mapped loci. We used $q$-value cutoff 0 (to include multi-mapped reads) and fragment size of 300. Due to the high correlation of replicate samples ($r^2 = 0.67$–$0.77$; $P$-value $< 2.2e$-16), two replicate reads were merged for the further analysis. After mapping of DamID-seq reads to the genome, $\log_2$ ratio of Lam-Dam/Dam was compared against Control- and Piwi- or Nxf2-KD according to the gene annotations downloaded from flybase (dmel-all-r5.57.gff).

To precisely analyze effect of Lamin DamID-seq signals against each type of TEs, genome-mapped reads were extracted and mapped to transposon consensus sequence obtained from Repbase (Jurka, 1998), permitting up to three mismatches but only those that mapped uniquely to the TE consensus sequence. In-house script was used in order to count GATC overlapped reads and calculate Lam-Dam/Dam $\log_2$ ratio, according to normalization obtained by genome-wide analysis using damidseq_pipleline. Dot plot and coverage of reads were generated using R.

LADs were defined as described previously (van Bemmel *et al*, 2010) with some modifications. Briefly, sharp transitions in the DamID signal were identified as a "border" region, and adjacent transitions were combined into domains if at least 75% of the enclosed probes have a positive $\log_2$ ratio. Each domain must have length of at least 5 kbp.

### ChIP-qPCR and ChIP-seq analysis

ChIP was performed as described previously (Iwasaki *et al*, 2016). Briefly, OSCs ($3 \times 10^7$) were fixed and lysed, and their nuclei were isolated using truChIP Chromatin Shearing Kits (Covaris), in accordance with the manufacturer's instructions. Sodium deoxycholate was added to a final concentration of 0.4% prior to shearing. DNA was sonicated to ˜300 bases using Bioruptor (Cosmobio) and then diluted with ChIP buffer [1 × 50 mM HEPES-KOH pH 7.4, 150 mM NaCl, 1 mM EDTA, 1% Triton X-100, Halt Protease Inhibitor Cocktail (Thermo Scientific)]. DNA–protein complexes were incubated with 3 μg of anti-H3K4me3 antibody (ab8580; Abcam) or H3K27Ac (ab4729; Abcam) on Dynabeads Protein G for 4 h, at 4°C. Beads were washed with ChIP buffer, high-salt lysis buffer (50 mM HEPES-KOH pH 7.4, 450 mM NaCl, 1 mM EDTA, 1% Triton X-100, 0.1% SDS), and wash buffer (50 mM Tris–HCl pH 8.0, 1 mM EDTA, 250 mM lithium chloride, 0.5% NP-40, 0.5% SDS) followed by TE (10 mM Tris–HCl pH 8.0, 1 mM EDTA). After reversing cross-linking for 12–16 h at 65°C, samples were treated with RNase for 30 min at 37°C and Proteinase K for 60 min at 55°C. DNA was then recovered.

To perform ChIP-qPCR, DNA templates were used to perform qPCR analysis as described in "Quantitative reverse-transcription PCR" section. Primers used for qPCR analysis are described in Table EV2. To prepare ChIP-seq libraries, DNA fragments from the ChIP experiment were sheared to ˜200 bases using Covaris S220 (Covaris). These were used for library preparation with the NEBNext Ultra II DNA Library Prep Kit for Illumina II (NEB), following the manufacturer's protocol. ChIP-seq libraries were sequenced with Illumina MiSeq using MiSeq Reagent Kit v3 for 150 cycles (Illumina).

### ChIP-seq data analysis

For ChIP-seq analysis, adapters were removed from the reads using Cutadapt, and subsequent reads shorter than 50 nt were discarded. ChIP-seq reads were mapped to the *Drosophila genome* (dm3) using Bowtie 2.2.9 (Langmead & Salzberg, 2012) with the default parameters. MACS 1.4.2 (Zhang *et al*, 2008) was used for peak calling and generation of wig files. Reads mapped to the genome were extracted and mapped to *Drosophila melanogaster* TE consensus sequences from Repbase (Jurka, 1998), permitting only those that mapped uniquely to the TE consensus sequence. Metaplots of ChIP signals were calculated using ngs.plot (Shen *et al*, 2014). Coordinates of TE insertions and the euchromatin/heterochromatin distribution within OSCs were obtained from a previous publication (Sienski *et al*, 2012; Data ref: Sienski *et al*, 2012). ChIP-seq data for H3K9me3 and H1 upon Piwi-KD were obtained from the previous study (Iwasaki *et al*, 2016; Data ref: Iwasaki *et al*, 2016).

### Oligo-FISH analysis

Oligo-FISH probe construction was performed as described previously (Beliveau *et al*, 2015; Beliveau *et al*, 2018). Oligo-FISH probes were designed as 26–32 nt target-site sequence flanked by 19 nt identical adaptors at both ends using OligoMiner (Beliveau *et al*, 2018). 80 sequences from each of 3–5 kb genomic sequence (chr2L:429,791–433,786; chr3L:17,550,003–17,554,855; chr3L:17,947,436–17,951,209) and 51 sequences from 1,921 bp luciferase reporter sequence were selected for oligo-FISH analysis. Oligonucleotide-probe sequences are listed in Table EV3. Oligonucleotide pools were amplified with 5′-fluorescence labeled (Cy3 or Cy5) and 5′-phosphorylated adaptor sequence primers using KOD One PCR Master Mix (TOYOBO), followed by purification using nucleospin Gel and PCR Clean-up (MACHEREY-NAGEL). The strands labeled Cy3 and phosphate were digested by Lambda Exonuclease (New England Biolabs) for 30 min at 37°C and then purified to obtain 5′-Cy3 labeled single-strand oligonucleotide probes. Oligo-FISH and immunofluorescence staining was performed as described previously (Markaki *et al*, 2013) with some modifications. Cells cultured on 18 × 18 mm coverslips were washed briefly with PBS, fixed with 4% formaldehyde/PBS for 10 min, rinsed with PBST (PBS + 0.02% Tween 20), quenched with 20 mM glycine/PBS for 10 min, and washed with PBST for 3 min. Next, samples were permeabilized with 1% Triton X-100/PBS for 10 min, washed with PBST 3-times, and incubated with 20% glycerol in PBS for 1 h.

Following that, samples were processed with freeze–thaw steps by dipping into liquid $N_2$ for 6-times. Samples were then washed with PBS for 2 min twice, incubated with 0.1 N HCl for 5 min, washed with PBS for 2 min twice and with 2× SSC (30 mM trisodium citrate dihydrate, 300 mM NaCl) for 2 min twice, and incubated with 50% formamide/2× SSC overnight at 4°C. Next day, samples were incubated with fresh 50% formamide/2× SSC for 30 min at 60°C. Probes were ethanol-precipitated with 10-times amounts of salmon sperm DNA (BioDynamics Laboratory) and dissolved into probe solution (50% formamide, 10% dextran sulfate, 2× SSC), denatured for 4 min at 86°C, and on ice for 2 min. Sample coverslips were mounted on a drop of 100 ng/5 μl probe mixture, heated for 3 min at 86°C, and then incubated in a floating box in a 37°C water bath for 3 days. After incubation, samples were washed with 2× SSC for 5 min at 37°C 3-times, with 0.1× SSC for 5 min at 60°C 3-times and with 4× SSCT (4× SSC + 0.02% Tween 20) for 1 min at 37°C 4-times. After oligo-FISH staining, samples were proceeded with immunofluorescence staining step; samples were treated with 2% BSA/PBST for 1 h, incubated with 2 μg/ml anti-Lam antibody (DHSB) /PBST for 1 h, washed with PBST for 5 min 3-times, incubated with Alexa Fluor488-conjugated anti-mouse IgG1 (Molecular Probes) /PBST (1:1,000 dilution) for 1 h, and washed with PBST 3-times. Samples were then post-fixed with 2% formaldehyde/PBST for 10 min, washed with PBST 3-times, counterstained with 2 μg/ml DAPI/PBST for 10 min, washed with PBST 3-times, and rinsed briefly with ddH$_2$O. Samples were mounted on a drop of VECTASHIELD Mounting Medium (Vector Laboratories). FISH and IF stained cells were observed using LSM 780 confocal microscopy (Carl Zeiss) with 63× Plan-Apochromat oil-immersion objective lens. 0.2-μm interval 30–40 z-section 3-color images were obtained, and chromatic shifts were corrected using Chromagnon (Kraus *et al*, 2017). Images were then processed with rescaling to 0.02 μm voxels and Gaussian blur (sigma = 3) using Fiji (Schindelin *et al*, 2012). Nuclear region was detected from DAPI images by thresholding, and DAPI center and DAPI surface were calculated. FISH spot center was also calculated from Cy3 images, and then, relative radial position of FISH spot against DAPI center was calculated. Statistical analysis, Mann–Whitney *U*-test, was performed using R.

# Data availability statement

The datasets produced in this study are available in the following databases:

Deep sequencing data: NCBI Gene Expression Omnibus GSE158082 (https://www.ncbi.nlm.nih.gov/geo/query/acc.cgi?acc = GSE158082).

Chromatin enrichment for proteomics data: Japan ProteOme STandard Repository JPST001092 (https://repository.jpostdb.org/entry/JPST001092).

Expanded View for this article is available online.

# Acknowledgements

We thank Shinsuke Shibata, Tomoko Shindo, Kensaku Murano, and Chikara Takeuchi for experimental support and Masaru Ariura for support and discussion on bioinformatic analyses. We also grateful to Dr. Bas van Steensel for sharing platforms for DamID experiments. We thank all the members of the Siomi laboratory, especially Hirotsugu Ishizu and Akihiko Sakashita for critical reading of the manuscript. Life Science Editors provided editing support on the manuscript. This work was supported by funding from JSPS KAKENHI Grant Numbers 19H05268 and 18H02421, JST PRESTO Grant Number JPMJPR20E2, The Uehara Memorial Foundation, Takeda Science Foundation, and The Nakajima Foundation to Y.W.I.; from JSPS KAKENHI Grant Number 19K23725 to Y.K.; from JSPS KAKENHI Grant Numbers 16H06279 and 19H04853 to W.I.; from JST CREST Grant Number JPMJCR18S3 to S.T.; and from JSPS KAKENHI Grant Number 19H05753 to H.S.

## Author contributions

YWI designed and performed most of the experiments and analyzed the data with AS. SS and WI analyzed Hi-C data. YK, YH, and ST performed oligo-FISH analysis. JA and TT performed ChEP analysis. YWI and HS conceived the study and wrote the paper with input from all the other authors.

## Conflict of interest

The authors declare that they have no conflict of interest.

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
