## [Review Process File · The EMBO Journal]

Piwi-piRNA complexes induce stepwise changes in nuclear architecture at target loci

Yuka Iwasaki, Sira Sriswasdi, Yasuha Kinugasa, Jun Adachi, Horikoshi Yasunori, Aoi Shibuya, Wataru Iwasaki, Satoshi Tashiro, Takeshi Tomonaga, and Haruhiko Siomi

DOI: [10.15252/embj.2021108345](https://doi.org/10.15252/embj.2021108345)

Corresponding author(s): Haruhiko Siomi (awa403@keio.jp), Yuka Iwasaki (iwasaki@keio.jp)

Review Timeline:

Submission Date:	26th Mar 21
Editorial Decision:	29th Apr 21
Revision Received:	30th May 21
Editorial Decision:	28th Jun 21
Revision Received:	4th Jul 21
Accepted:	9th Jul 21

Editor: Stefanie Boehm

Transaction Report:

(Note: Depending on transfer agreements, referee reports obtained elsewhere may or may not be included in this compilation. Referee reports are anonymous unless the Referee chooses to sign their reports.)

29th Apr 2021

Re: EMBOJ-2021-108345

Piwi-piRNA complexes induce stepwise changes in nuclear architecture at target loci

Dear Dr. Siomi,

Thank you for submitting your manuscript to The EMBO Journal. Please excuse the delay in communicating this decision you, which was due to repeatedly delayed referee reports. We have meanwhile received two reports on your study (included below for your information) and decided to make a preliminary decision based on these. Given the two referees' comments, we would like to invite you to start preparing a revised manuscript. Should referee #2's report arrive in due time, I will forward the comments and contact you to discuss how to best proceed.

As you will see, both reviewers appreciate the analysis and quality of the study. However, they do raise some concerns and questions that should be addressed in the revised version. Referee #1 brings up a current question in the field, if nuclear position controls transcription or if transcription affects position (point 1). Please add to the discussion on this aspect as the referee suggests and also address the issue experimentally, if technically feasible (point 2). This may also help to address the concerns of referee #3 regarding the effect of localization on silencing. Please also experimentally address referee #2's questions regarding potential effects Piwi depletion has on the cell cycle and viability, or include/discuss prior data on this aspect. In addition, please also address all technical concerns both referees indicate by providing additional information on methodology, adding statistical analyses and revising the text accordingly (ref#1- point 3, 4; 5 ref#3- point 4 (Fig 5), 5, 9). Moreover, please remember to provide public access to all datasets of the study (ref#3- point 8) and appropriately revise the text upon carefully considering all other referee comments in your point-by-point response.

Please note that it is our policy to allow only a single round of major revision. We realize that lab work worldwide is currently affected by the COVID-19/SARS-CoV-2 pandemic and that an experimental revision may be delayed. If you foresee any potential issues that may significantly delay a revision, please contact us to discuss this. Please also do not hesitate to contact us if you have any other questions or would like to discuss a revision plan in more detail.

Thank you again for the opportunity to consider your work for publication. I look forward to receiving your revised manuscript!

Kind regards,

Stefanie Boehm

Stefanie Boehm
Editor
The EMBO Journal

Referee #1:

This is very carefully conducted study with interesting results. The main findings are that Piwi together with its partner protein Nxf2 controls the spatial positioning of some transposable elements in *Drosophila* overy cells. Both positioning relative to the lamina (detected by DamID-seq and FISH) and relative to other genomic loci (detected by Hi-C) are affected. The role of egg, HP1 and histone H1 are also studied in detail. The quality of the data appears to be consistently very high, the experimental designs are well-controlled, the data analyses have been done with great care, and the figures are generally beautiful and easily interpretable. I consider this paper thus of very high quality. I have a few minor comments:

1. A general conundrum in the field is whether nuclear positioning controls gene expression or vice versa (chicken vs egg problem). There is some evidence that transcription can counteract lamina interactions and modulate Hi-C contacts. The relocation of the Piwi targets after Piwi or Nxf2 KD could thus be a secondary effect of the transcription activation - I recommend that this possibility is discussed more explicitly in the Discussion section. However, a handful of papers have also shown that depletion of lamins can lead to activation of repressed, lamina-associated genes. But these latter findings are still rare. In this light, figure EV1C is particularly interesting. Although I agree with the authors that the magnitude of the derepression is modest, I do think that this result is important enough to be shown in a main figure rather than in an EV figure. It may also be good to highlight this result a bit more in the text, as it provides useful insights into the chicken-egg problem, and it also implicates the lamina as another player besides egg, HP1, and H1.
2. In connection to the previous point: Perhaps the authors could attempt a double-KD of lam and lamC to see if this would lead to stronger derepression?
3. Figures 2B/5B: please clarify whether the sequence in the gypsy peak around 940k is sufficiently unique that DamID-seq reads can indeed be uniquely mapped there. Or is this peak the result of reads derived from other gypsy elements that are cross-mapped to this locus?
4. Figure 3B-C: the units of the color key (-100..100, resp -500..500) are not meaningful. Please replace by % change or fold-change. Same for Fig 3I and 4A. Figs 3DEF would also be biologically more interpretable if z-scores were replaced by fold-change or % change. Same for Figure 6A,C-E.
5. Fig 5A: please provide evidence of the reproducibility of these data: quantification of band intensities across independent biological replicates.
6. p16: "These results suggest a temporal model where Piwi-piRNA first affects nuclear positioning of LADs and active histone marks, and this is followed by regulation of TADs and repressive histone marks." -- I could not find the evidence indicating that Piwi-piRNA **first** affects nuclear positioning, as no DamID or FISH data are presented for this tethering experiment?

=====Bas van Steensel, review assignment accepted: 28 March 2021, completed 31 March 2021. It is my standard policy to sign *all* of my reviews, irrespective of my comments and recommendations. All correspondence regarding this manuscript should go via the editor. [PLEASE DO NOT REMOVE THIS NOTE]=====

Referee #3:

In this manuscript, Iwasaki and colleagues set out to characterize the process of piRNA-guided heterochromatin formation and transposon silencing in *Drosophila* from the following angles:

1. Does piRNA-guided silencing result in or correlate with changes of nuclear localization of targeted loci?
2. Does piRNA-guided silencing result in or correlate with changes of chromosome conformation or chromosomal contacts at targeted loci?
3. Is there a temporal order in distinct molecular steps that contribute to silencing?

The authors conclude that:

1. piRNA-guided transcriptional repression leads to an increased association of targeted loci with the nuclear lamina (by performing Lamin-DamID and FISH experiments)
2. piRNA-guided transcriptional silencing leads to subtle yet consistent changes in intra TAD and inter TAD interactions (by performing Hi-C and FISH experiments)
3. piRNA-guided transcriptional silencing displays a temporal order of 'removal of active chromatin marks and change in localization' followed by 'transcriptional silencing' and 'changes in chromosome conformation' followed by 'establishment of heterochromatin'.

Overall, this manuscript addresses, using a cell culture system and siRNA mediated knockdowns, a number of questions that are per se of interest in the field. The presented data are overall of high quality. In terms of impact and advancing our understanding of the molecular mechanisms at place, the findings are mostly descriptive. This is especially true when the authors attempt to reconcile their findings of Lamina association with the Nxf2 protein that acts downstream of the Piwi-piRNA complex.

Prior to this work, it was established that Piwi induces strong transcriptional silencing and heterochromatin formation at its target loci. Moreover, the Siomi labs have previously shown that silencing leads to reduced chromatin accessibility (by ATAC-seq) and is accompanied by increased incorporation of Histone 1 at targeted loci. Considering this, it is maybe not too surprising that depletion of Piwi leads to changes in active chromatin marks (H3K27ac and H3K4me) and changes in chromosomal contacts at target loci. The change in nuclear localization on the other hand is a novel finding. Here, however, I am not fully convinced whether the presented data is sufficiently conclusive and whether targeting of the locus to the Lamina is relevant for silencing. All in all, I have a few specific comments to the presented experiments. Considering the overall impact, I fear that I am not convinced that the presented data presents an important step forward in our understanding of how piRNA-guided transcriptional silencing operates at the molecular level.

To the HiC experiments: Though I am not an expert in this area, the presented data in terms of quality and analysis seems solid. The overall impact of the findings is limited in my opinion given the previous findings that Piwi-mediated silencing has such strong impacts on transposon silencing and heterochromatin formation. Loss of Piwi will lead to a substantial change in the transcription profile

of the targeted loci. Not unexpectedly, this leads to changes in intra-TAD and inter-TAD interactions. It is assuring to see that this is the case, but the insight gained from these data, at least at first glance, seems rather limited.

On the Lamina-association and re-localization of target loci to the nuclear envelope: The authors start out with a ChEP experiment and find that Piwi loss leads to a globally detectable reduction in proteins of the nuclear Lamina to crosslink to Chromatin. While this is consistent with their follow-up experiments (FISH), it is actually rather surprising. The fraction of the genome or even transcriptome that is targeted by Piwi for silencing is small. It is therefore very surprising to see a measurable change by mass-spec in the association between chromatin and lamin-proteins at a global scale. Given that no direct physical or genetic link between the nuclear lamina and Piwi or the downstream PICTS complex could be documented, I do wonder whether the observation of reduced chromatin-lamina association could not be a consequence of Piwi-depleted cells accumulating in a different cell cycle stage (as demonstrated by the authors) or due to these cells being impacted in their overall viability (Piwi-depleted cells start to stop proliferating and dying after 4 days of siRNA mediated knockdown).

The Dam-ID experiments using lamin as a fusion protein are convincing. They indicate that at steady state, piRNA targeted loci have a relatively higher tendency to be lamina associated compared to expressed control loci. And that loss of Piwi leads to a dis-engagement of these loci from the lamina. Association of heterochromatic areas with the lamina is a known phenomenon. It is therefore good and interesting to see that this is also true for piRNA loci. The impact on piRNA-mediated silencing is, however, rather unclear. Loss of lamin or laminC via siRNA-mediated depletion leads to very mild defects in transposon repression, indicating that lamina association is not required for efficient silencing. The authors also show changes in nuclear localization of target loci by FISH experiments (Figure 5). It is rather difficult to interpret panel L. The IF image shown looks extremely convincing, but the quantification of this trend is much less clear to me. According to this quantification, there is only a weak trend that Piwi-targeting of a locus results in association of the locus with the nuclear lamina.

On a technical note: Did the authors sequence the genome of the OSC line or how were the transposon insertions mapped and annotated? Also, for the screenshots showing the Dam-ID data: Are these showing genome-unique reads or all mappers? There are dozens of gypsy insertions in the OSC genome and it is surprising to see so many (unique?) mappings to the gypsy insertion.

Citations: The authors do carefully cite the literature. Here and there, I recommend to double-check the citations. Yu et al. is for example not cited when it comes to the involvement of HP1 or SetDB1 as Piwi downstream factors.

Intro: It was a bit surprising that the authors introduce Nxf2-Nxt1 as a sort of stand-alone complex rather than explaining right away that Nxf2 is part of a multi-protein complex that has even been characterized by the Siomi lab.

The list of the 25 identified proteins in the ChEP experiment, or ideally the entire dataset would be important to publish alongside.

Figure 3G: Without a control locus, this is difficult to judge for me. The change in distance is (maybe expected) very small. Is this specific to piRNA target loci?

Referee #1:

This is very carefully conducted study with interesting results. The main findings are that Piwi together with its partner protein Nxf2 controls the spatial positioning of some transposable elements in *Drosophila* overy cells. Both positioning relative to the lamina (detected by DamID-seq and FISH) and relative to other genomic loci (detected by Hi-C) are affected. The role of egg, HP1 and histone H1 are also studied in detail. The quality of the data appears to be consistently very high, the experimental designs are well-controlled, the data analyses have been done with great care, and the figures are generally beautiful and easily interpretable. I consider this paper thus of very high quality. I have a few minor comments:

Thank you for your positive comments. We also appreciate the constructive comments, which have helped us greatly improve our manuscript.

1. A general conundrum in the field is whether nuclear positioning controls gene expression or vice versa (chicken vs egg problem). There is some evidence that transcription can counteract lamina interactions and modulate Hi-C contacts. The relocation of the Piwi targets after Piwi or Nxf2 KD could thus be a secondary effect of the transcription activation - I recommend that this possibility is discussed more explicitly in the Discussion section. However, a handful of papers have also shown that depletion of lamins can lead to activation of repressed, lamina-associated genes. But these latter findings are still rare. In this light, figure EV1C is particularly interesting. Although I agree with the authors that the magnitude of the derepression is modest, I do think that this result is important enough to be shown in a main figure rather than in an EV figure. It may also be good to highlight this result a bit more in the text, as it provides useful insights into the chicken-egg problem, and it also implicates the lamina as another player besides egg, HP1, and H1.

Thank you for addressing this important point. We added a paragraph regarding this in the discussion section, which now reads as follows: "A general conundrum in this field is whether nuclear positioning controls gene expression or vice versa (Luperchio et al, 2014; van Steensel & Furlong, 2019). There is some evidence that transcription can counteract lamina interactions and modulate Hi-C contacts. The relocation of piRNA target TEs after Piwi- or Nxf2-KD could be a secondary effect of transcriptional activation. On the other hand, slight de-repression of piRNA target TEs upon the depletion of lamin proteins (Fig 1E) may suggest a possible defect in transcription upon disruption of lamina association. Defects in transcriptional regulation and nuclear localization upon Piwi-KD occur prior to changes in H3K9me3 histone marks and chromatin conformation (Fig 4, 5L) (Murano et al., 2019). In *C. elegans*, heterochromatin regions

with H3K9me3 histone marks are anchored to perinuclear regions (Towbin et al, 2012). This suggests that in the Piwi-piRNA pathway, deposition of H3K9me3 at a later step reinforces the association of chromatin regions with perinuclear regions. Indeed, Piwi or Nxf2 knockdown, which decreases the Lamin DamID-seq signal at Piwi-piRNA target regions (Fig 2, 5), also results in a major decrease in H3K9me3 levels (Batki et al., 2019; Fabry et al., 2019; Murano et al., 2019; Zhao et al., 2019). These results suggest that lamina localization may both be the cause and result of TE de-repression. Nuclear peripheral localization may affect gene expression to some extent, and the regulation of gene expression may be enforced due to its localization to the nuclear periphery.”

As suggested, we have moved old Fig EV1C to Fig 1E (new) and modified the text, which now reads as follows: “However, Lam- or LamC-KD resulted in a slight de-repression of piRNA target TEs (Fig 1E), suggesting that accession to the nuclear periphery may impact TE repression to some extent. Due to severe damage to Lam- or LamC-KD cells, we could only perform knockdown for 48 hours, which cannot completely deplete their expression (Fig 1C). Similarly, double KD of Lam and LamC proteins severely affected cell viability. Incomplete KD may result in the underestimation of its effect on TE de-repression upon lamin loss.” We have also used these data (Fig 1E) to discuss the “chicken-egg problem,” as described above.

2. In connection to the previous point: Perhaps the authors could attempt a double-KD of lam and lamC to see if this would lead to stronger derepression?

Thank you for your suggestion. We attempted double-KD of Lam and LamC, but this killed most of the cells (~96% of double-KD cells died compared to control-KD), and consequently, we could not collect sufficient cells to perform qPCR analysis. Even if we could collect enough cells by scaling-up the experiment, we are afraid that the surviving cells may reflect an unusual cell population. This was consistent with the fact that Lam- or LamC-KD can only be performed in 48 h, which results in incomplete depletion of their expression (in the case of Piwi-KD, we performed KD in 96 h by transfecting siRNAs twice). Severe knockdown of lamina proteins appears to affect cell viability adversely, making it difficult to discuss their impact. We have also included this aspect in the text (described in response to comment #1).

3. Figures 2B/5B: please clarify whether the sequence in the gypsy peak around 940k is sufficiently unique that DamID-seq reads can indeed be uniquely mapped there. Or is this peak the result of reads derived from other gypsy elements that are cross-mapped to this locus?

This figure is generated from reads mapped using bowtie2 with default parameters, which accepts reads that are multi-mapped (cross-mapped). However, the multi-mapped reads were randomly mapped to only one of the best-mapped loci; therefore, they were not overcounted. Since it is difficult to distinguish the same type of transposon in different genomic positions, we also performed analyses to re-map the genome mapped reads to transposon consensus sequences and discuss changes in each type of transposon (Figure 2D-F and Figure 5C-E). As suggested, we have clarified this point in the legends of Figures 2B/5B and added the description to the materials and methods section.

4. Figure 3B-C: the units of the color key (-100..100, resp -500..500) are not meaningful. Please replace by % change or fold-change. Same for Fig 3I and 4A. Figs 3DEF would also be biologically more interpretable if z-scores were replaced by fold-change or % change. Same for Figure 6A,C-E.

We thank the reviewer for highlighting this. We used \log_2 fold-change to generate the heatmaps shown in Figures 3B-C, 3I, 4A, 5G, and 6A (new). This clarified the increase in long-range interactions. Also, by using \log_2 fold-change, we could not observe a slight non-specific increase of short-range interactions upon Piwi- or Nxf2-KD (we hypothesized that it is likely to be due to the global change in cell condition upon KD). Therefore, we amended the manuscript by deleting the corresponding text. This does not alter the conclusion of the paper.

We preferred the use of z-scores to analyze the association between changes in TAD-level Hi-C contact and TE insert regions (Figure 3D-F, 6C-E) because we wanted to remove biases in Hi-C changes that might be specific to each chromosome or TAD size. Our method converts the fold change in Hi-C contact for each TAD (or a pair of TADs) to a z-score by normalizing it against the mean and standard deviation of fold changes in Hi-C contact in a random locus (or a pair of loci) with the same size(s) as the TAD and on the same chromosome. Hence, using z-scores here provides a fair comparison of Hi-C changes across TADs of different sizes and from different chromosomes.

5. Fig 5A: please provide evidence of the reproducibility of these data: quantification of band intensities across independent biological replicates.

We quantified band intensities across three independent biological replicates and showed the reproducibility of these data (shown in new Fig EV4A).

6. p16: "These results suggest a temporal model where Piwi-piRNA first affects nuclear positioning of LADs and active histone marks, and this is followed by regulation of TADs and repressive histone marks." -- I could not find the evidence indicating that Piwi-piRNA *first* affects nuclear positioning, as no DamID or FISH data are presented for this tethering experiment?

We proposed the temporal model here, since active histone marks are affected at the earlier (48 h) time point (Figure 4E), and Lamin-DamID signals, but not TADs, were correlated with changes in active histone marks (Figure 4C,D). We also described FISH data for the tethering experiment (in time course) in Figure 5L. However, as the figure comes after this sentence, it may confuse the readers. In addition, *first* might be an overstatement. We therefore removed "first" from this sentence and added the following text for clarity: "Together with the correlation of LADs with active histone marks and TADs with repressive histone marks (Fig 4C, 4D; see also Fig 5L), these results suggest a temporal model where Piwi-piRNA affects nuclear positioning of LADs and active histone marks, and this is followed by regulation of TADs and repressive histone marks."

=====Bas van Steensel, review assignment accepted: 28 March 2021, completed 31 March 2021. It is my standard policy to sign *all* of my reviews, irrespective of my comments and recommendations. All correspondence regarding this manuscript should go via the editor. [PLEASE DO NOT REMOVE THIS NOTE]=====

We thank Dr. Bas van Steensel for the valuable comments. We also appreciate the DamID method, which enabled us to perform a series of experiments to reveal the relationship between nuclear localization and Piwi-piRNA regulation.

Referee #2:

No report from Referee #2

Referee #3:

In this manuscript, Iwasaki and colleagues set out to characterize the process of piRNA-guided heterochromatin formation and transposon silencing in Drosophila from the following angles:

- 1. Does piRNA-guided silencing result in or correlate with changes of nuclear localization of targeted loci?
- 2. Does piRNA-guided silencing result in or correlate with changes of chromosome conformation

or chromosomal contacts at targeted loci?

3. Is there a temporal order in distinct molecular steps that contribute to silencing?

The authors conclude that:

1. piRNA-guided transcriptional repression leads to an increased association of targeted loci with the nuclear lamina (by performing Lamin-DamID and FISH experiments)
2. piRNA-guided transcriptional silencing leads to subtle yet consistent changes in intra TAD and inter TAD interactions (by performing Hi-C and FISH experiments)
3. piRNA-guided transcriptional silencing displays a temporal order of 'removal of active chromatin marks and change in localization' followed by 'transcriptional silencing' and 'changes in chromosome conformation' followed by 'establishment of heterochromatin'.

Overall, this manuscript addresses, using a cell culture system and siRNA mediated knockdowns, a number of questions that are per se of interest in the field. The presented data are overall of high quality. In terms of impact and advancing our understanding of the molecular mechanisms at place, the findings are mostly descriptive. This is especially true when the authors attempt to reconcile their findings of Lamina association with the Nxf2 protein that acts downstream of the Piwi-piRNA complex.

Thank you for summarizing our findings and indicating that we have addressed several questions that are of interest in this field.

Prior to this work, it was established that Piwi induces strong transcriptional silencing and heterochromatin formation at its target loci. Moreover, the Siomi labs have previously shown that silencing leads to reduced chromatin accessibility (by ATAC-seq) and is accompanied by increased incorporation of Histone 1 at targeted loci. Considering this, it is maybe not too surprising that depletion of Piwi leads to changes in active chromatin marks (H3K27ac and H3K4me) and changes in chromosomal contacts at target loci. The change in nuclear localization on the other hand is a novel finding. Here, however, I am not fully convinced whether the presented data is sufficiently conclusive and whether targeting of the locus to the Lamina is relevant for silencing. All in all, I have a few specific comments to the presented experiments. Considering the overall impact, I fear that I am not convinced that the presented data presents an important step forward in our understanding of how piRNA-guided transcriptional silencing operates at the molecular level.

Thank you for your valuable comments. Whether nuclear positioning controls gene expression or vice versa is a general conundrum in the field and is very difficult to answer, as referee #1 mentioned in the first comment (see also response to referee #1 comment #1). We included a discussion of this point in the revised manuscript, indicating the possibility that relocation of the piRNA target TEs after Piwi or Nxf2 knockdown could be a secondary effect of transcriptional activation. The revised text reads as follows: “A general conundrum in this field is whether nuclear positioning controls gene expression or vice versa (Luperchio et al, 2014; van Steensel & Furlong, 2019). There is some evidence that transcription can counteract lamina interactions and modulate Hi-C contacts. The relocation of piRNA target TEs after Piwi- or Nxf2-KD could be a secondary effect of transcriptional activation. On the other hand, slight de-repression of piRNA target TEs upon the depletion of lamin proteins (Fig 1E) may suggest a possible defect in transcription upon disruption of lamina association. Defects in transcriptional regulation and nuclear localization upon Piwi-KD occur prior to changes in H3K9me3 histone marks and chromatin conformation (Fig 4, 5 L) (Murano et al., 2019). In *C. elegans*, heterochromatin regions with H3K9me3 histone marks are anchored to perinuclear regions (Towbin et al, 2012). This suggests that in the Piwi-piRNA pathway, deposition of H3K9me3 at a later step reinforces the association of chromatin regions with perinuclear regions. Indeed, Piwi or Nxf2 knockdown, which decreases the Lamin DamID-seq signal at Piwi-piRNA target regions (Fig 2, 5), also results in a major decrease in H3K9me3 levels (Batki et al., 2019; Fabry et al., 2019; Murano et al., 2019; Zhao et al., 2019). These results suggest that lamina localization may both be the cause and result of TE de-repression. Nuclear peripheral localization may affect gene expression to some extent, and the regulation of gene expression may be enforced due to its localization to the nuclear periphery.”

Most of the presented data may be descriptive, but we believe that this manuscript describes an important step forward in our understanding of the mechanism through which piRNA-guided transcriptional silencing operates. Extending our previous study on the Piwi-Panx-Nxf2-p15 (PPNP) complexes (Murano et al., *EMBOJ*, 2019) and changes in chromatin accessibility (Iwasaki et al., *Mol Cell*, 2016), we now show that PIWI-piRNA complexes can induce not only local changes in histone modifications at the target loci but also impact genome-wide spatial changes by using Hi-C and Lamin-DamID approaches. Since it has only been described that the Piwi-piRNA system can affect the local and neighboring chromatin states, this study presents an important step forward in terms of its impact on genome-wide nuclear architecture. Moreover, by analyzing the link between changes in different histone marks, nuclear localization at the periphery, and chromatin conformation, we demonstrated a hierarchy between these regulations. The strengths of this study are that it connects our emerging understanding of Panx-Nxf2-Nxt1 in

the piRNA pathway to lamins, chromatin architecture, and gene expression regulated at the nuclear periphery. Altogether, the findings of this study provide insights into the temporal steps of silencing involving localization followed by removal or addition of chromatin modifications, which provides a broader conceptual understanding that connects piRNA-mediated silencing and general heterochromatin formation mechanism.

To the HiC experiments: Though I am not an expert in this area, the presented data in terms of quality and analysis seems solid. The overall impact of the findings is limited in my opinion given the previous findings that Piwi-mediated silencing has such strong impacts on transposon silencing and heterochromatin formation. Loss of Piwi will lead to a substantial change in the transcription profile of the targeted loci. Not unexpectedly, this leads to changes in intra-TAD and inter-TAD interactions. It is assuring to see that this is the case, but the insight gained from these data, at least at first glance, seems rather limited.

Thank you for your valuable comment. We believe that it is important to show that the Piwi-piRNA system can affect not only the local chromatin state but also the nuclear architecture of nuclear localization, as shown in this study. Additionally, the analysis using Piwi-KD in OSC tended to underestimate the impact of Piwi depletion on the 3D structure of the genome. This is because the population of piRNA target TEs is smaller in OSCs than in flies. It is also known that chromatin conformation stabilizes after its formation, and the impact can be observed to be markedly severe in the context of developmentally regulated events (Szabo et al., *Sci Adv*, 2019). Another reason could be the fact that we did not use highly affected cells (which died) in our study.

To estimate the impact of Piwi loss *in vivo*, we performed Hi-C experiments using ovaries from Piwi mutant *Drosophila piwi* [1/ Δ 37] (Jin et al., *Current Biol*, 2013; Zallen et al., *J Cell Biol*, 2002; Cox et al., *Genes Dev*, 1998). By comparing the Piwi mutant versus the wild type, we observed a major decrease in short-range interactions and an increase in long-range interactions (log₂ FC heatmap (A) and contacts per distance (B) are shown below). The comparison of these data to Piwi-KD in OSCs (Figure 3B and Figure EV2B in the revised manuscript) indicates that the *piwi* mutant has a similar effect on the 3D structure but on a much larger scale. Since we did not have information on TE insertion in Piwi mutant flies, we could not perform further analysis to describe the impact of Piwi loss in flies precisely. In addition, since Piwi mutant *Drosophila* ovaries have major morphological defects, we did not include these data in the revised manuscript. However, these data suggest that the impact of Piwi-KD in OSCs can be observed on a markedly larger scale in Piwi mutant flies.

We added this point in the discussion section, which now reads as follows: “Since *piwi* mutant *Drosophila* has severe ovary morphological defect (Cox et al, 1998), our study was performed using a knockdown-based cell line system. However, it is likely that the impact of chromatin conformation would be larger in *piwi* mutant *Drosophila* than in the cell line. This is because the population of piRNA target TEs is smaller in OSCs, and also because we did not use highly affected cells upon Piwi-KD in our study, as they died (Fig EV5E). Additionally, it is known that the chromatin conformation is highly stable once formed, and the impact is more severe for developmentally regulated events (Szabo et al, 2019). Therefore, it would be both interesting and important to observe the *in vivo* impact of Piwi-piRNA regulation on chromatin conformation in the future.”

On the Lamina-association and re-localization of target loci to the nuclear envelope: The authors start out with a ChEP experiment and find that Piwi loss leads to a globally detectable reduction in proteins of the nuclear Lamina to crosslink to Chromatin. While this is consistent with their follow-up experiments (FISH), it is actually rather surprising. The fraction of the genome or even transcriptome that is targeted by Piwi for silencing is small. It is therefore very surprising to see a measurable change by mass-spec in the association between chromatin and lamin-proteins at a global scale. Given that no direct physical or genetic link between the nuclear lamina and Piwi or the downstream PICTS complex could be documented, I do wonder whether the observation of reduced chromatin-lamina association could not be a consequence of Piwi-depleted cells accumulating in a different cell cycle stage (as demonstrated by the authors) or due to these cells being impacted in their overall viability (Piwi-depleted cells start to stop proliferating and dying after 4 days of siRNA mediated knockdown).

Thank you for your comment. Since piRNA target TEs are distributed over different regions of the genome and their impact tends to spread to tens of kbs of neighboring regions, we thought it is possible that they could have genome-wide effects detectable by ChEP analysis. Accordingly,

we performed a series of follow-up experiments to confirm this. We have added the following sentence to the revised text for clarification: “It is unclear whether the decrease in the association of chromatin with lamins, as indicated by ChEP analysis, is linked to Piwi-piRNA regulation or global changes in cellular conditions upon Piwi-KD.”

Please note that all analyses described in this manuscript were performed after removing dead cells, which are suspected to have a marked effect on cell cycle and proliferation. This was in line with the impact of Piwi-KD on the cell cycle, which was limited (Figure EV5B). We amended the sentence to clarify this point which now reads as follows: “For HP1a-KD, we found a decrease in populations at the G1 state and increase in populations at the G2-M state, where the impact on cell cycle upon Piwi-KD was limited.” We also included this point in the discussion section (see response to the comment for “To the HiC experiments”).

The Dam-ID experiments using lamin as a fusion protein are convincing. They indicate that at steady state, piRNA targeted loci have a relatively higher tendency to be lamina associated compared to expressed control loci. And that loss of Piwi leads to a dis-engagement of these loci from the lamina. Association of heterochromatic areas with the lamina is a known phenomenon. It is therefore good and interesting to see that this is also true for piRNA loci. The impact on piRNA-mediated silencing is, however, rather unclear. Loss of lamin or laminC via siRNA-mediated depletion leads to very mild defects in transposon repression, indicating that lamina association is not required for efficient silencing. The authors also show changes in nuclear localization of target loci by FISH experiments (Figure 5). It is rather difficult to interpret panel L. The IF image shown looks extremely convincing, but the quantification of this trend is much less clear to me. According to this quantification, there is only a weak trend that Piwi-targeting of a locus results in association of the locus with the nuclear lamina.

Although the association of heterochromatic areas with the lamina is a known phenomenon, as piRNA target TEs are also located at the non-heterochromatic areas (Sienski et al., *Cell*, 2012), it would be interesting to investigate their localization. Importantly, we further analyzed the effect of Piwi-KD using two experiments (DamID and FISH), and both suggested changes in nuclear localization.

As the cells that were markedly affected by Piwi-KD died, we did not include these populations in our analyses. The exclusion of dead cells likely results in an underestimation of the impact of Piwi depletion, and it could be the reason why the observed changes were smaller. This point is discussed in the revised text (see also the response to comment “To the HiC experiments:”).

Similar observations were made for Lamin or LaminC, whose knockdown results in even more severe defects in cells. We could only perform a 2-day knockdown, which does not completely repress the expression of lamina proteins (4-day knockdown kills most of the cells), and double knockdown of Lamin and LaminC resulted in severe defects in the cells (see response to referee #1 comment #2). Therefore, we could not evaluate the effect of complete lamina protein depletion, which may have resulted in an underestimation of the effect on de-repression of TEs.

As suggested by the reviewer, we observed DamID signal at representative piRNA loci in OSCs (*flam* and *tj-3'UTR*) and observed the association of *flam* with lamina but not *tj*. Thank you for your valuable comment. However, since this would lead to a different study, we would further investigate this aspect after the current study is published.

On a technical note: Did the authors sequence the genome of the OSC line or how were the transposon insertions mapped and annotated? Also, for the screenshots showing the Dam-ID data: Are these showing genome-unique reads or all mappers? There are dozens of gypsy insertions in the OSC genome and it is surprising to see so many (unique?) mappings to the gypsy insertion.

We used the *Drosophila* reference genome, but the positions for transposon insertions were obtained from a previous publication that sequenced the OSC genome to identify their transposon insertion (Sienski et al., *Cell*, 2012). OSCs were gifted to the Brennecke group from our lab; we speculate that the transposon insertion sites described by their sequencing should

not considerably differ from that of our OSCs. We have added this to the materials and methods section.

We included multi-mapped reads for the mapping shown in Figures 2B and 5B, but further analysis was performed by re-mapping the genome-mapped reads to the TE consensus sequence. We clarified this in the legends of Figures 2B and 5B and the materials and methods section (also see response to referee #1 comment #3).

Citations: The authors do carefully cite the literature. Here and there, I recommend to double-check the citations. Yu et al. is for example not cited when it comes to the involvement of HP1 or SetDB1 as Piwi downstream factors.

Thank you for your comment. We checked the citation and added Yu et al. at the part describing the involvement of HP1 or SetDB1 as Piwi downstream factors.

Intro: It was a bit surprising that the authors introduce Nxf2-Nxt1 as a sort of stand-alone complex rather than explaining right away that Nxf2 is part of a multi-protein complex that has even been characterized by the Siomi lab.

Thank you for your suggestion. We have added a sentence describing the multi-protein complex before introducing Nxf2-Nxf1. The revised text reads as follows: "Recently, it has been identified that the Piwi-Panx-Nxf2-p15 (PPNP, Pandas, PICTS, or SFiNX) complex, plays a central role in bridging target-associated Piwi-piRNA to chromatin regulation and transcriptional silencing (Batki et al, 2019; Fabry et al, 2019; Murano et al, 2019; Zhao et al, 2019)."

The list of the 25 identified proteins in the ChEP experiment, or ideally the entire dataset would be important to publish alongside.

Thank you for your suggestion. We added a table describing the 25 proteins identified in the ChEP experiment (new Table EV1). Additionally, we uploaded our raw proteome data to a database (Japan ProteOme STandard Repository: accession# JPST001092) for further analysis.

Figure 3G: Without a control locus, this is difficult to judge for me. The change in distance is (maybe expected) very small. Is this specific to piRNA target loci?

The change in the distance is expected to be small based on the change in intra-TAD interactions, as reported by a previous study (Szabo Q et al., *Sci Adv.*, 2018). It is very difficult to identify a suitable “control locus” to describe whether the described change in the distance is adequately large. Therefore, we described that there was a significant difference between control (EGFP)- and Piwi-KD at the same loci using Oligo-FISH to confirm the results of the HiC analysis. To show that the change in intra-TAD interactions is specific to piRNA target loci, we used HiC analysis (Figure 3) instead of Oligo-FISH.

28th Jun 2021

Re: EMBOJ-2021-108345R

Piwi-piRNA complexes induce stepwise changes in nuclear architecture at target loci

Dear Haru,

Thank you for submitting your revised manuscript we have now received the reports from the two initial referees (see comments below). I am pleased to say that they overall find that their comments have been satisfactorily addressed and in principle now support publication. Nonetheless, both referees point out some minor issues that should be resolved in a final textual revision of the manuscript. Please also provide a brief point-by-point response to these comments. In addition, I would like to ask you to address a number of editorial issues listed in detail below. Please make any changes to the manuscript text in the attached document only using the "track changes" option. Once these remaining issues are resolved, we will be happy to formally accept the manuscript for publication.

Thank you again for giving us the chance to consider your manuscript for The EMBO Journal. I look forward to receiving your final revision. Please feel free to contact me if you have further questions regarding the revision or any of the specific points listed below.

Kind regards,
Stefanie

Stefanie Boehm
Editor
The EMBO Journal

Referee #1:

The authors have further improved and clarified the manuscript. I recommend publication, but I would still urge the authors to clarify how the z-scores in Figs 3D-F, 6C-E translate into fold-change; at least a broad indication of the effect sizes would help: is it 1.1-fold, 2-fold, 5-fold?

Please also note that Figure 4A seems still not in log₂ fold change units, unlike what is mentioned in the rebuttal.

=====
Bas van Steensel, review assignment accepted: 2 June 2021, completed 2 June 2021. It is my standard policy to sign **all** of my reviews, irrespective of my comments and recommendations. All correspondence regarding this manuscript should go via the editor. [PLEASE DO NOT REMOVE THIS NOTE]=====

Referee #3:

In their revised manuscript, H. Siomi and colleagues carefully addressed the various comments, suggestions and critiques by the reviewers. By adding aspects to the discussion section and more details in the experimental section, this led to an improved and more balanced presentation of the data. I do agree with reviewer #1 that the quality of the experimental data and the overall analysis are at a high level. While I am still not entirely convinced that the data represent a major step forward in our mechanistic understanding of the Piwi-mediated silencing process, I agree that the presented data and the manuscript overall report insights of interest to the field.

I have some minor comments for the authors:

1. In their response to both reviewers (e.g. page 2, line 6 of the author response letter), it is stated: "These results suggest that lamina localization may both be the cause and result of TE de-repression." If I am not mistaken, it should read "TE-repression" instead.

2. On the comment that piwi mutant ovaries show an even more pronounced change in chromatin conformation: "To estimate the impact of Piwi loss in vivo, we performed Hi-C experiments using ovaries from Piwi mutant *Drosophila piwi* [1/Δ37] (Jin et al., *Current Biol*, 2013; Zallen et al., *J Cell Biol*, 2002; Cox et al., *Genes Dev*, 1998). By comparing the Piwi mutant versus the wild type, we observed a major decrease in short-range interactions and an increase in long-range interactions (log₂ FC heatmap (A) and contacts per distance (B) are shown below). The comparison of these data to Piwi-KD in OSCs (Figure 3B and Figure EV2B in the revised manuscript) indicates that the piwi mutant has a similar effect on the 3D structure but on a much larger scale. Since we did not have information on TE insertion in Piwi mutant flies, we could not perform further analysis to describe the impact of Piwi loss in flies precisely. In addition, since Piwi mutant *Drosophila* ovaries have major morphological defects, we did not include these data in the revised manuscript. However, these data suggest that the impact of Piwi-KD in OSCs can be observed on a markedly larger scale in Piwi mutant flies."

A key issue I see with comparing HiC maps of wildtype versus piwi mutant ovaries is that in the mutant, ovarian development is grossly altered (also stated by the authors). This is expected to lead to a strong reduction in the number of nurse and follicle cells that underwent endo-replication. As a consequence, one might expect the observed changes in HiC maps simply because of this. In early stages of fly oogenesis, the endoreplicated chromatids are even bundled together in polytene-like chromosomes, which should lead to a strong increase in short range interactions by HiC measurements. I therefore recommend deleting even the reduced comment on this experiment in the manuscript.

3. On the multi-mapping analysis that was also raised by reviewer #1. The best approach in my opinion is to analyze all mappers for the analysis on TE consensus sequences (as done by the authors), but to restrict the analysis to genome-unique mappings when it comes to the analysis of genomic loci harboring single, stand-alone TE insertions. This would shift the focus on reads that immediately flank the TE insertion and would give strong support to a model that also the stand-alone TE insertions in general euchromatin are closer to the lamina if they are repressed by the piRNA pathway. It would be ideal if the authors include an analysis like this where they focus on the DamID signal in the regions immediately flanking piRNA-repressed TE insertions in wildtype and Piwi-depleted OSCs.

Referee #1:

The authors have further improved and clarified the manuscript. I recommend publication, but I would still urge the authors to clarify how the z-scores in Figs 3D-F, 6C-E translate into fold-change; at least a broad indication of the effect sizes would help: is it 1.1-fold, 2-fold, 5-fold?

Thank you for the comments. We believe that our manuscript has been greatly improved by the inputs from the Referees.

Regarding the interpretation of the Z-score, the exact conversion will depend on the dataset. Each Z-score can be converted to a fold-change using the standard deviation and average values. Please note that the reported average and standard deviations are on the Log₂ scale, so the conversion formula is $\text{fold-difference} = 2^{(Z\text{-score} \times \text{std} + \text{average})}$.

In the case of the EGFP-KD vs. Piwi-KD dataset in this manuscript, rough estimates for the conversion of Z-score to the fold difference is:

For intra-TAD analysis, one unit of Z-score is ~0.02 unit of fold difference.

For inter-TAD analysis, one unit of Z-score is ~0.16 unit of fold difference.

We also added the following sentence to the manuscript to further clarify this part. "Then, the Z-score for each intra- and inter-TAD interaction was calculated using the formula $Z\text{-score} = (\log_2 \text{fold change} - \text{average } \log_2 \text{fold change}) / \text{standard deviation of } \log_2 \text{fold change}$."

Please also note that Figure 4A seems still not in log₂ fold change units, unlike what is mentioned in the rebuttal.

Thank you very much for pointing this out. Although we have updated the heatmap for Figure 4A, the scale bar has not been updated. We amended Figure 4A.

=====
Bas van Steensel, review assignment accepted: 2 June 2021, completed 2 June 2021. It is my standard policy to sign *all* of my reviews, irrespective of my comments and recommendations. All correspondence regarding this manuscript should go via the editor.
[PLEASE DO NOT REMOVE THIS NOTE]=====

Referee #2:

No report from Referee #2

Referee #3:

In their revised manuscript, H. Siomi and colleagues carefully addressed the various comments, suggestions and critiques by the reviewers. By adding aspects to the discussion section and more details in the experimental section, this led to an improved and more balanced presentation of the data. I do agree with reviewer #1 that the quality of the experimental data and the overall analysis are at a high level. While I am still not entirely convinced that the data represent a major step forward in our mechanistic understanding of the Piwi-mediated silencing process, I agree that the presented data and the manuscript overall report insights of interest to the field.

Thank you for pointing out that our manuscript overall reports insights of interest to the field. We believe that the manuscript has been greatly improved by the inputs from the Referees.

I have some minor comments for the authors:

1. In their response to both reviewers (e.g. page 2, line 6 of the author response letter), it is stated: "These results suggest that lamina localization may both be the cause and result of TE de-repression." If I am not mistaken, it should read "TE-repression" instead.

Thank you very much. This should have read "TE-repression," as pointed out by the reviewer. We have amended the corresponding text within the manuscript, which now reads: "These results suggest that lamina localization may both be the cause and result of TE repression."

2. On the comment that piwi mutant ovaries show an even more pronounced change in chromatin conformation: "To estimate the impact of Piwi loss in vivo, we performed Hi-C experiments using ovaries from Piwi mutant *Drosophila piwi* [1/Δ37] (Jin et al., Current Biol, 2013; Zallen et al., J Cell Biol, 2002; Cox et al., Genes Dev, 1998). By comparing the Piwi mutant versus the wild type, we observed a major decrease in short-range interactions and an increase in long-range interactions (log₂ FC heatmap (A) and contacts per distance (B) are shown below). The comparison of these data to Piwi-KD in OSCs (Figure 3B and Figure EV2B in the revised manuscript) indicates that the piwi mutant has a similar effect on the 3D structure but on a much

larger scale. Since we did not have information on TE insertion in Piwi mutant flies, we could not perform further analysis to describe the impact of Piwi loss in flies precisely. In addition, since Piwi mutant *Drosophila* ovaries have major morphological defects, we did not include these data in the revised manuscript. However, these data suggest that the impact of Piwi-KD in OSCs can be observed on a markedly larger scale in Piwi mutant flies."

A key issue I see with comparing HiC maps of wildtype versus piwi mutant ovaries is that in the mutant, ovarian development is grossly altered (also stated by the authors). This is expected to lead to a strong reduction in the number of nurse and follicle cells that underwent endo-replication. As a consequence, one might expect the observed changes in HiC maps simply because of this. In early stages of fly oogenesis, the endoreplicated chromatids are even bundled together in polytene-like chromosomes, which should lead to a strong increase in short range interactions by HiC measurements. I therefore recommend deleting even the reduced comment on this experiment in the manuscript.

Thank you for the suggestion. We have deleted the comment on this experiment, as suggested by the Referee. The deleted section reads: "Since piwi mutant *Drosophila* has severe ovary morphological defect (Cox et al, 1998), our study was performed using knockdown-based cell line system. However, it is likely that the impact of chromatin conformation would be larger in piwi mutant *Drosophila* than in the cell line. This is because the population of piRNA target TEs is smaller in OSCs, and also because we did not use highly affected cells upon Piwi-KD in our study, as they died (Fig EV5E). Additionally, it is known that the chromatin conformation is highly stable once formed, and the impact is more severe for developmentally regulated events (Szabo et al, 2019). Therefore, it would be both interesting and important to observe the in vivo impact of Piwi-piRNA regulation on chromatin conformation in future." Also, since this is the only part that refers to Fig EV5E, we have also deleted Fig EV5E (which shows the ratio of alive cells upon Piwi-KD).

3. On the multi-mapping analysis that was also raised by reviewer #1. The best approach in my opinion is to analyze all mappers for the analysis on TE consensus sequences (as done by the authors), but to restrict the analysis to genome-unique mappings when it comes to the analysis of genomic loci harboring single, stand-alone TE insertions. This would shift the focus on reads that immediately flank the TE insertion and would give strong support to a model that also the stand-alone TE insertions in general euchromatin are closer to the lamina if they are repressed by the piRNA pathway. It would be ideal if the authors include an analysis like this where they focus on the DamID signal in the regions immediately flanking piRNA-repressed TE insertions in wildtype and Piwi-depleted OSCs.

Thank you for the important comment. Our DamID read distribution shown in Figure 4A would be a good example of a region immediately flanking piRNA-repressed TE insertions in Control- vs. Piwi-KD OSCs, pointed out by the reviewer. By looking at this region, we can see that the effect of lamina association affects not only TE insertion itself but also the flanking regions. We added a sentence to describe this, which reads: “Importantly, we could observe that the changes in Hi-C contacts and DamID-seq signals also spread to the neighboring regions of TE insertion (Figure 4A).”

Thank you again for submitting the final revised version of your manuscript. I am pleased to inform you that we have now formally accepted it for publication in The EMBO Journal.

Corresponding Author Name: Yuka W. Iwasaki, Haruhiko Siomi

Journal Submitted to: The EMBO Journal

Manuscript Number: EMBOJ-2021-108345